# Seasonality and climate modes influence the temporal clustering of unique atmospheric rivers in the Western U.S
Zhiqi Yang [1] ✉, Michael J. DeFlorio[1], Agniv Sengupta [1], Jiabao Wang[1], Christopher M. Castellano[1],
Alexander Gershunov [1], Kristen Guirguis[1], Emily Slinskey[1], Bin Guan [2,3], Luca Delle Monache[1] &
F. Martin Ralph[1]

Atmospheric rivers (ARs) are narrow corridors of intense water vapor transport, shaping precipitation, floods, and economies. Temporal clustering of ARs tripled losses compared to isolated events, yet the reasons behind this clustering remain unclear. AR orientation further modulates hydrological impacts through terrain interaction. Here we identify unique ARs over the North Pacific and Western U.S. and utilize Cox regression and composite analysis to examine how six major climate modes influence temporal clustering of unique ARs and orientation during extended boreal winter (November to March). Results show that climate modes condition temporal clustering of unique ARs. The Pacific-North American weather pattern strongly modulates the clustering over the Western U.S. from early to late winter. The quasi-biennial oscillation and Pacific decadal oscillation affect late winter clustering, while the Arctic oscillation dominates early winter. Climate modes also strongly influence AR orientation, with ENSO particularly affecting the orientation of temporally clustered ARs.

Atmospheric rivers (ARs) are characterized by intense horizontal water vapor transport within a narrow corridor, profoundly shaping precipitation, floods, water availability, ecosystems, and economies. Specifically, ARs significantly contribute to flood risks and precipitation in the Western U.S.[1–3], and flood damages exhibit exponential growth with the intensity and duration of AR occurrences[1]. When multiple ARs occur in rapid succession, the compound effect on the hydrologic system can lead to more hazards[4], more impacts on the ecosystem[5], and more economic losses[1]. AR sequences are also aligned with periods of increased hydrologic hazards in the historical record, such as extreme precipitation, landslides, and flooding[4]. Also, the orientation of ARs further influences the severity of their impacts, as whether AR hits the terrain more perpendicularly can significantly affect hydrological outcomes[6]. Under climate change, the frequency of ARs has the potential to increase in the future[7], suggesting the increasing likelihood and importance of temporal clustering of ARs. The occurrence of nine consecutive atmospheric rivers within a mere three-week span in California during winter 2022/2023 highlighted the temporal AR clustering feature, garnering increased attention[8]. Bowers et al.[9] made a striking discovery: temporal clustered ARs lead to over three times higher expected losses in California compared to individual ARs occurrence.

This underscores the need to understand the temporal compounding and orientation aspects for precise prediction of AR impacts.

Several previous studies have been conducted to explore the characteristics of AR sequences that occur closely together in time, also referred to as "AR families" or "temporal clustering"[10–13]. Here, we consider unique AR events (in short, unique ARs); for a unique AR event, if it occurred on consecutive days, only the first day is counted. Unique AR events exhibiting temporal clustering generally show alternation between quiet and active periods regarding the number of events. This differs from events that happen independently of each other, as seen in memoryless Poisson processes (see Methods). The occurrence of nine consecutive ARs within a three-week span in California during winter 2022/2023 serves as an illustration of temporal AR clustering. Fish et al.[11] introduced the concept of AR families, identifying periods when multiple ARs occurred in rapid succession in Northern California. They found that half of all ARs belong to AR families and that some of these clusters are strongly linked to the El Niño/Southern Oscillation (ENSO) and the Madden-Julian Oscillation (MJO). Slinskey et al.[12] examined subseasonal AR clustering in the Western U.S., showing ARs temporally cluster at a higher than random rate and investigating the features of extreme precipitation coincident with temporally clustered ARs, but the underlying physical mechanisms are still unclear.

[1]Center for Western Weather and Water Extremes, Scripps Institution of Oceanography, University of California San Diego, San Diego, CA, USA. [2]Joint Institute for Regional Earth System Science and Engineering, University of California Los Angeles, Los Angeles, CA, USA. [3]Jet Propulsion Laboratory, California Institute of Technology, Pasadena, CA, USA. ✉e-mail: zhy040@ucsd.edu

Zhou et al.[13] studied temporal AR clusters and their hydrological impacts in the Western U.S., finding a significant link to a Pacific-North American (PNA)-like pattern. Previous research has also explored temporal clustering in other hydrometeorological variables, such as streamflow in the eastern U.S., heavy precipitation, and hurricanes in the Central U.S., Europe, and China[5,14–19] as well as the ability of reanalysis data and climate models (GCM/RCM) to capture these clustering characteristics[19–21]. These studies highlight the significant influence of large-scale atmospheric circulation on the temporal clustering of hydrometeorological variables. Yet, the enigma persists: what is the interacting effect of climate modes and seasonality on AR temporal clustering? Answering this question is crucial as it will yield new approaches and predictors for improving the forecasting of temporal AR clustering and related economic losses at subseasonal-to-seasonal timescales.

Moreover, the orientation of ARs has distinct effects on the intensity and duration of their associated precipitation and floods, with particular emphasis on their orientation upon landfall due to the interplay between moisture transport and complex terrain. Slinskey et al.[22] found that AR integrated water vapor transport (IVT) along the U.S. West Coast typically has a southwesterly orientation, aligning with moisture pathways from the subtropics to the extratropics. Guirguis et al.[23] demonstrated that persistent southerly IVT orientations contributed to the exceptionally wet winter of 2016–2017 in Northern California, linking these ARs to significant rainfall. South-southwesterly ARs are associated with more precipitation and stronger moisture transport in Northern California, while westerly orientations are linked to extreme streamflow and intense precipitation in Washington and Oregon[24,6,25]. In other regions, west-southwesterly orientations yield the most pronounced hydrological responses in Britain's Dyfi catchment[26]. Climate modes like ENSO can also modulate AR orientation at landfall[23]. At subseasonal lead times, AR orientation is crucial, as some orientations are much more predictable in the Western U.S. and have significant impact implications[27]. This critical insight underscores the need for targeted forecasting strategies that account for these variations in predictability, making it an essential consideration in hydrological predictions and risk assessments. Therefore, certain AR orientations can worsen flood severity, especially when they occur in quick succession. So, what is the interacting effect of climate modes and seasonality on AR orientation? Addressing this question is essential, as it will enhance the forecasting of AR orientation. When combined with improved forecasting of temporal AR clustering, this will allow for more accurate estimation of subsequent hydrological impacts and benefit water management efforts in the Western U.S.

Motivated by these two questions, this study delves into this research gap, concentrating on the temporal clustering of unique ARs in the Western U.S. during boreal winter (November to March), highlighting the role of dominant climate modes in modulating the temporal clustering of ARs and AR orientation. Since climate modes are important predictors to subseasonal-to-seasonal (S2S) forecasting modeling studies, here we consider several major climate modes with important impacts on the Western U.S., including the Arctic Oscillation (AO)[28], quasi-biennial oscillation (QBO)[29], ENSO, MJO[30,31], Pacific Decadal Oscillation (PDO)[32], and Pacific-North American pattern (PNA)[33,34]. These climate modes can modulate meteorological parameters crucial to the Western U.S., such as IVT, precipitation[23,35], AR duration, frequency, landfall orientation[23,35,36], and snow water equivalent (SWE)[37]. Moreover, since seasonal disparities exist in the modulations of ARs by climate modes, we further investigate the differences in clustering characteristics between early winter and late winter[35,36].

Here, we find that the temporal clustering of unique ARs during boreal winter is conditioned on the occurrence of the climate modes. The PNA significantly influences unique AR temporal clustering in the Western U.S., with QBO and PDO affecting late winter clustering and AO dominating early winter, according to analysis of six climate modes' influence on unique AR clustering. Climate modes also strongly influence AR orientation, such as ENSO, affecting the orientation of temporally clustered unique ARs. The discovery of relationships between factors with high subseasonal

predictability and AR clustering/orientation would lay a foundation to evaluate the subseasonal (2–6 week lead) and seasonal (3–6 month lead) predictability of AR sequences occurring within a short time period. Water resource managers and other applied end users across the Western U.S. stand to reap numerous benefits, such as minimization of flood risks and enhanced planning and decision-making, from improved predictability of hydroclimate phenomena at these extended lead times[8,38–40].

## Results

### Temporal AR clustering in the winter season—November to March

To understand the average number of unique ARs (see Methods) in the 1-week window over the Western U.S., we first compare the climatological patterns of unique ARs in extended boreal winter (November to March; NDJFM), early winter (November-December-January; NDJ), and late winter (January-February-March; JFM) from 1982 to 2021 in Fig. 1. The regions with the maximum unique AR numbers are in the Central and Western North Pacific, with around 1.4 unique ARs per week. Near the Western U.S., unique ARs decrease from the coastline (to around 1 ARs per week) towards the inland region (to around 0.6 ARs per week). Therefore, the climatology for the inland region has less unique ARs than the climatology for the coast.

Then, we investigate the temporal clustering of unique ARs within the extended boreal winter over the Western U.S., and the impact of AO, QBO, ENSO, MJO, PDO, and PNA (see Methods) on the magnitude and spatial structure of the clustering. Figure 2, left column, shows how climate modes influence temporal AR clustering during the extended boreal winter (NDJFM). Statistically significant Cox regression coefficients $\beta$ (shaded areas) indicate that unique ARs exhibit temporal clustering. The Cox process belongs to the family of clustered processes, indicating the occurrence of one event has connections with the subsequent one. $\beta$ represents the regression coefficient for each covariate, quantifying the influence of climate modes on the cluster distribution of unique ARs, where higher $|\beta|$ values indicate stronger temporal AR clustering (see Methods). Note that PDO and the other five modes are analyzed in separate regressions, making a direct comparison of PDO coefficients with the second regression not possible (see Methods). Specifically, the AO influences Northern California, with more unique AR clustering during its negative phase, consistent with Guan et al.[41]. The QBO's positive phase increases unique AR clustering in the Southwestern U.S. from California to New Mexico, while its negative phase reduces it. ENSO impacts are mainly over the ocean, with weak effects on land. The MJO's RMM1 and RMM2 show distinct patterns, with RMM1 indicating a north-positive/south-negative pattern and RMM2 significant in the Central U.S[42] (see Methods). The impacts of the PDO are mostly seen as a dipole pattern from Washington/Oregon to the Rocky Mountains and California. Positive PDO phases increase unique AR clustering over California, while decreasing it from Washington/Oregon to the Rockies; negative PDO has the opposite effect, aligning with Gershunov et al.[43], demonstrating that the PDO significantly modulates AR frequency over West Coast. Note that both the PDO and AO significantly impact regions from the North Pacific to the West Coast, warranting further investigation into their interactions in future studies. The PNA signal shows a northwest-southeast pattern over the Western U.S. The PNA's positive phase increases unique AR clustering in the Northwestern U.S., while its negative phase does so in the interior Southwestern U.S., as noted by Zhou et al.[13] in investigating PNA-like pattern related to AR clustering. PNA shows a strong impact as it is a mode of climate variability affecting Western North America, particularly affecting the Western U.S. Therefore, these findings highlight how different climate modes modulate AR temporal clustering. By incorporating the influence of large-scale climate modes and considering temporal clustering, they enable more accurate S2S forecasting of AR events (similar example: hurricane forecast improvement[44]).

To quantitatively measure the number of unique ARs associated with each phase of these climate modes during the extended boreal winter season (NDJFM), we consider the composite analysis of the number of unique ARs

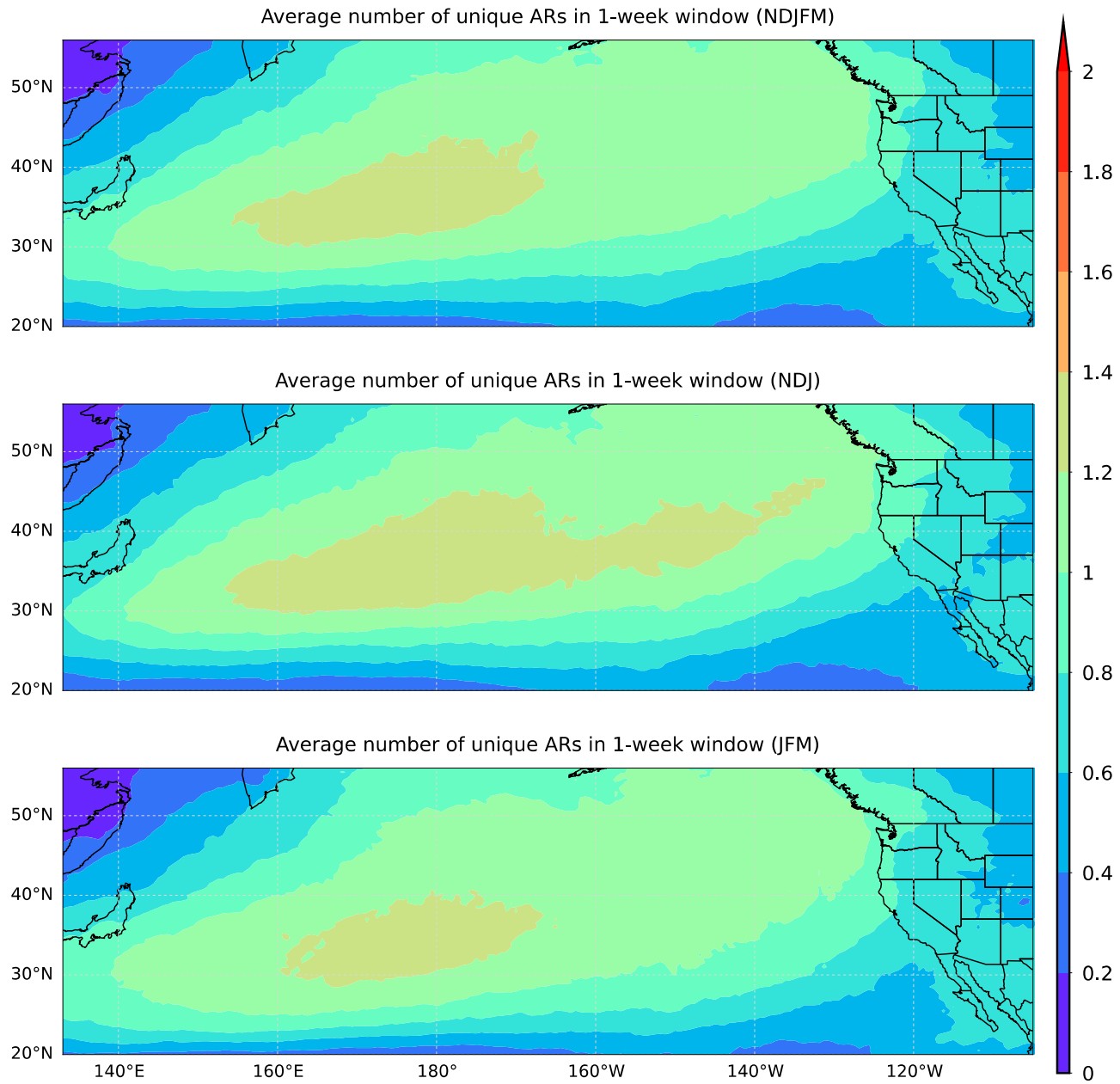

**Fig. 1 | Average number of unique ARs in a 1-week window.** Average number of unique ARs in a 1-week window based on the MERRA-2 AR dataset from 1982/1983 to 2020/2021 during extended winter (NDJFM, top panel), early winter (NDJ, middle panel), and late winter (JFM, bottom panel). Unit: number of unique ARs.

anomalies in a 1-week window (see Methods) based on the phase of AO, QBO, ENSO, MJO, PDO, and PNA in Fig. 3. Figure 3 spatial patterns highly match the Cox regression coefficients patterns in Fig. 2 left column, supporting that unique ARs exhibit temporal clustering and are affected by climate modes. Specifically, during the AO's negative phase, unique ARs in Northern California increase by about 0.1 per week compared to climatology. In the QBO's positive phase, unique ARs in the Southwestern U.S. rise by 0.05–0.1 ARs per week compared to climatology. ENSO shows weak effects in the Western U.S. Positive PDO conditions correspond to significantly higher AR temporal clustering over California by around 0.01 ~ 0.05 ARs per week, while negative PDO conditions correspond to significantly higher AR temporal clustering from Washington/Oregon to the Rocky Mountains by around 0.05–0.1 ARs per week. Positive PNA conditions correspond to higher AR temporal clustering in the Northwestern U.S. by around 0.2 ARs per week; negative PNA conditions correspond to higher AR temporal clustering in the interior Southwestern U.S. by about 0.2–0.3 ARs per week. These results offer an independent approach

from the previous paragraph, highlighting how climate modes modulate temporal AR clustering and showing the variation in the number of unique ARs associated with the positive and negative phases of climate modes within a one-week window. Consequently, we can improve prediction of the number and impact of ARs based on the current phase of these climate modes.

To understand the influence of the MJO more clearly, we consider the composite analysis of the number of unique ARs over a 1-week window based on four combined MJO phases (consider amplitude>1) at a lag time of five days: MJO phases 1 and 8 (enhanced convection over the Western Hemisphere and Africa), phases 2 and 3 (Indian Ocean), phases 4 and 5 (Maritime Continent), and phases 6 and 7 (Western Pacific)[42]. Supplementary Fig. 1 shows the composite analysis of unique ARs based on MJO phase, and Fig. 4 shows the anomalies during the extended boreal winter season (NDJFM). AR temporal clustering shows significant modulation across all MJO phases. Specifically, Phases 1 and 8 correspond to about 0.1 fewer unique ARs per week across most of the Western U.S. than

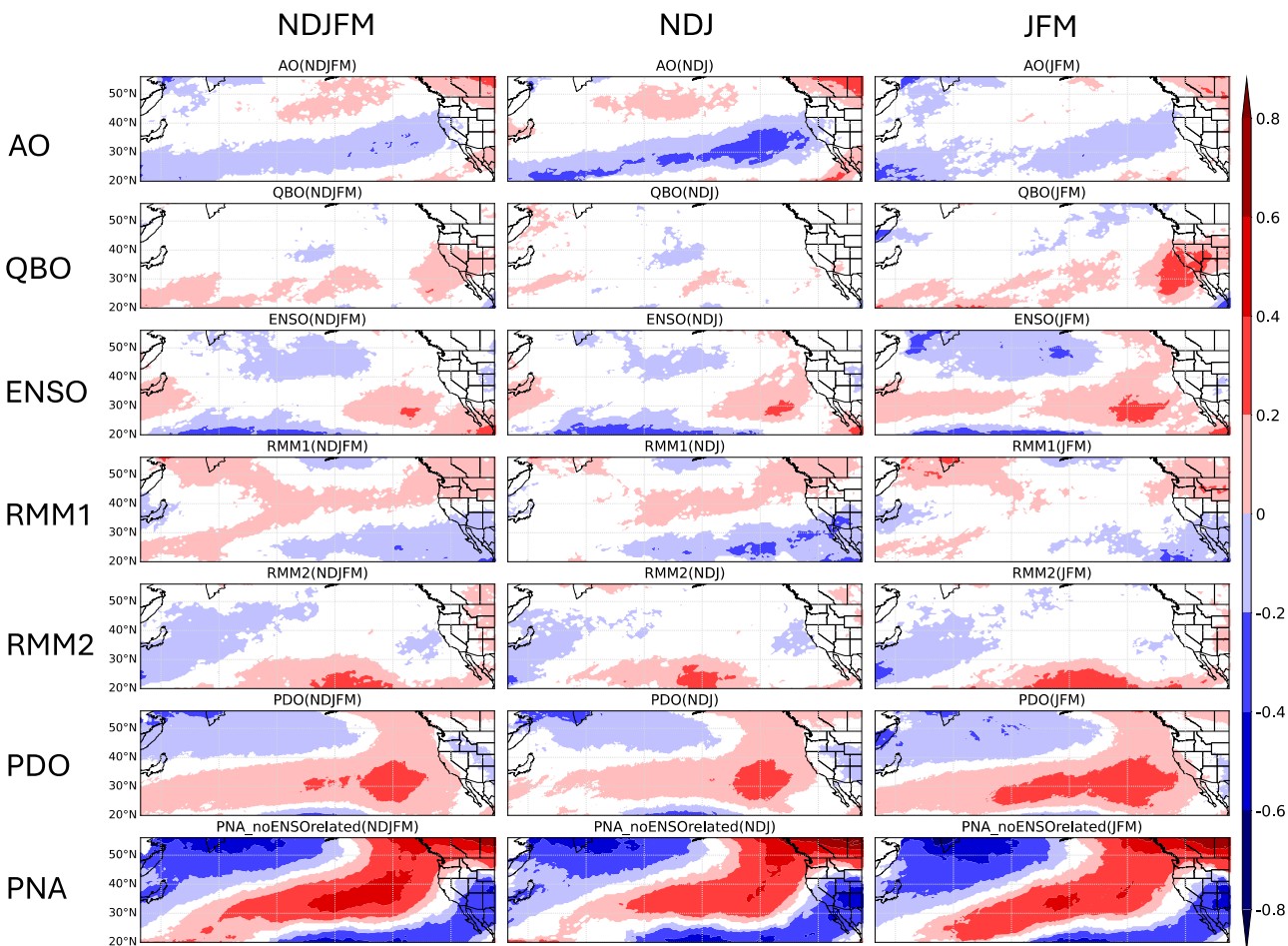

**Fig. 2 | Cox regression coefficients showing climate modes modulations on temporal clustering of unique ARs.** Cox regression coefficients between AO, QBO, ENSO, MJO-RMM1, MJO-RMM2, PDO, PNA (removed ENSO signal), and the occurrence of unique ARs based on the ERA-5 AR dataset from 1940/1941 to 2017/2018 (for PDO analysis) and MERRA-2 AR dataset from 1982/1983 to 2020/2021 (for other climate modes analysis) during extended winter (NDJFM, left column), early winter (NDJ, middle column), and late winter (JFM, right column). Note that the coefficients from the regression of AO, QBO, ENSO, MJO, and PNA cannot directly be comparable to those from PDO (see Methods). The results show coefficients that are statistically significant at the 5% level.

climatology. In contrast, phases 4 and 5 increase unique ARs in the Northwestern U.S. and western Canada but decrease them in the Southwestern U.S., particularly Southern California, by about 0.2 ARs per week. Phases 2 and 3 decrease AR clustering in the Southwestern U.S., while phases 6 and 7 increase it, especially in Southern California, which is consistent with previous studies[45]. Since MJO phases are an important source of S2S predictability, this finding offers valuable insights into the expected number of ARs based on the current MJO phase, thereby improving forecast accuracy.

### Seasonality of temporal AR clustering during extended boreal winter

In addition to the influence of climate modes on the temporal clustering of unique atmospheric rivers during extended boreal winter, it is also important to consider the seasonality of this temporal clustering[35]. We now consider temporal AR clustering during the early (NDJ) vs. late (JFM) winter periods over the North Pacific and Western U.S. region. To address this inquiry, Fig. 2 middle and right columns illustrate the Cox regression coefficient $\beta$ concerning AO, QBO, ENSO, MJO, PDO, and PNA during NDJ (middle) and JFM (right) (see Methods). The clustering patterns vary significantly from early to late winter, suggesting strong seasonality of temporal AR clustering and its modulation by modes of climate variability during early vs. late winter. Specifically, the AO mainly impacts Northern California in early winter, while the QBO's influence is stronger in late winter in the Southwestern U.S. and relatively small during early winter,

which is consistent with the results from Castellano et al.[36]. ENSO and PDO effects are more pronounced in late winter, with ENSO's influence along the California coastline and PDO's influence stronger over the ocean and California, and in early winter from Washington/Oregon to the Rocky Mountains. When PDO is positive, temporal AR clustering increases in California during late winter and decreases in Washington/Oregon and the Rocky Mountains during early winter, with opposite effects in its negative phase. The main difference in the influence of the PNA during early vs. late winter can be seen with positive values of $\beta$ extending further south in the equatorial Pacific in late winter. Therefore, this better understanding of how climate modes like AO, QBO, ENSO, MJO, PDO, and PNA influence AR clustering differently in early winter (NDJ) versus late winter (JFM) allows us to tailor AR predictions based on the specific characteristics and behaviors of these climate modes during different parts of the winter season.

We now quantify the difference in the impact of climate mode variability in modulating temporal AR clustering between early vs. late winter. Figure 5 shows composites in anomalies of the number of unique ARs in a 1-week window based on the phase of AO, QBO, ENSO, MJO, PDO, and PNA for early vs. late winter; anomalies mean difference compared to climatology. In general, the composite analysis aligns with the Cox regression $\beta$ patterns in Fig. 2 (middle and right), confirming the seasonality of AR clustering. When the AO is in its negative phase, California sees about 0.1 more unique ARs per week in early winter, relative to climatology (Fig. 5a). During the QBO's positive phase, the Southwestern U.S., especially California, experiences ~0.1 more unique

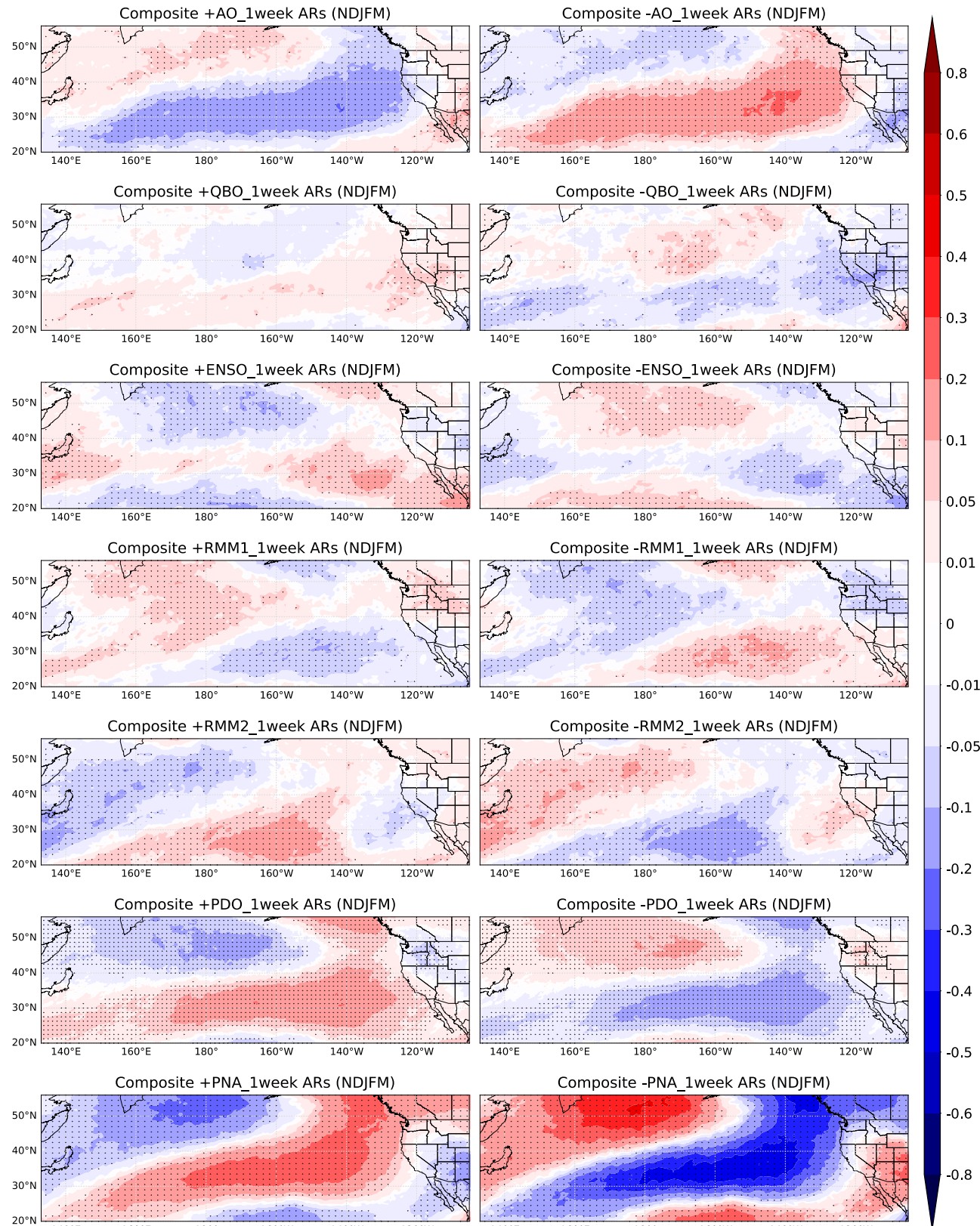

**Fig. 3 | Composite analysis of anomalies in the number of unique ARs based on the positive and negative phases of climate modes during extended winter.** Composite analysis in anomalies of the number of unique ARs in 1-week window during extended winter (NDJFM) based on positive (left panels) and negative phases (right panels) of AO, QBO, ENSO, MJO-RMM1, MJO-RMM2, PDO, PNA (removed ENSO signal), using the ERA-5 AR dataset from 1940/1941 to 2017/2018 (for PDO analysis) and MERRA-2 AR dataset from 1982/1983 to 2020/2021 (for other climate modes analysis). Dots indicate statistical significance at the 5% level. Unit: number of unique ARs.

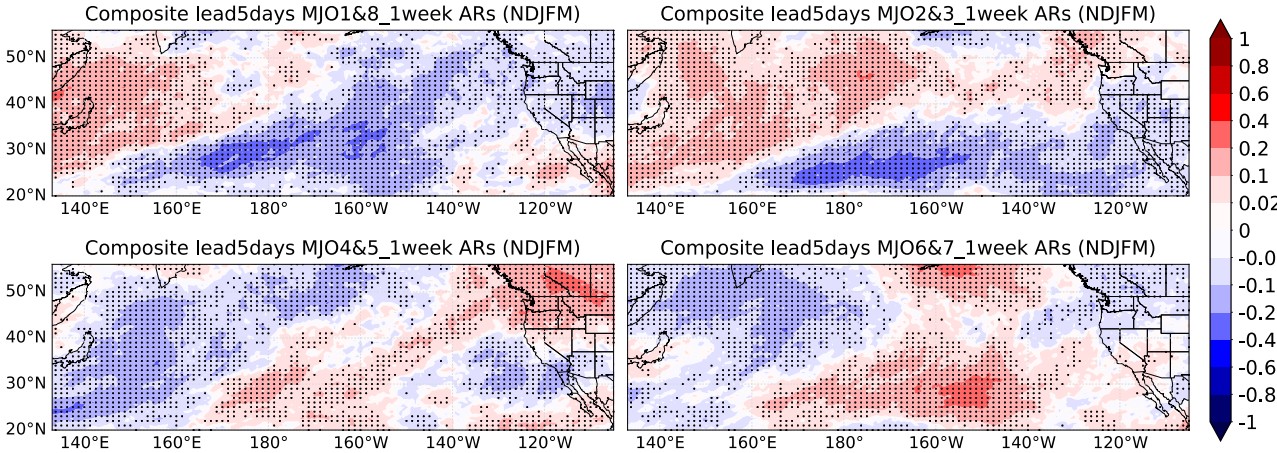

**Fig. 4 | Composite analysis of anomalies in the number of unique ARs based on four combined MJO phases during extended winter.** Composite analysis in anomalies of the number of unique ARs in a 1-week window during extended winter (NDJFM) based on four combined MJO phases using the MERRA-2 AR dataset from 1982/1983 to 2020/2021. Dots indicate statistical significance at the 5% level. Unit: number of unique ARs.

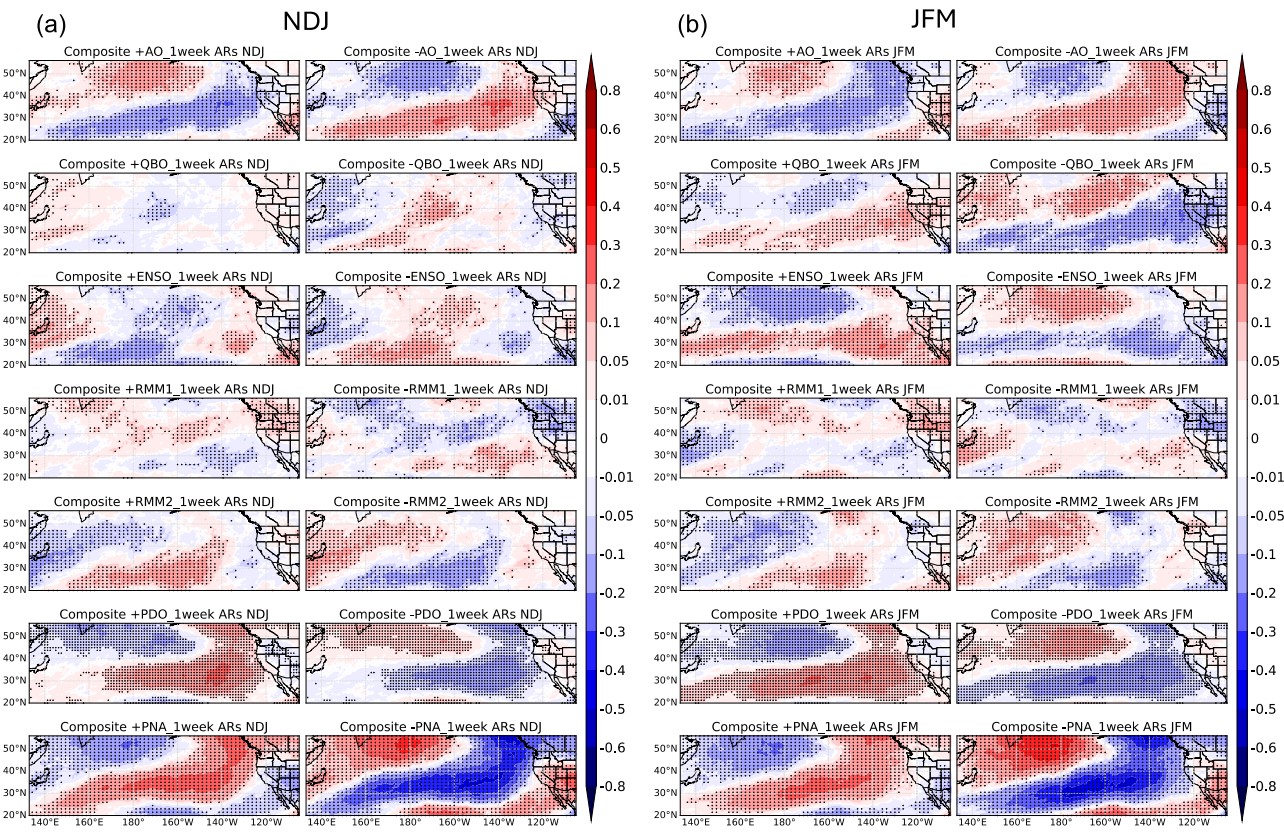

**Fig. 5 | Composite analysis of anomalies in the number of unique ARs based on six climate modes during early and late winter.** Similar to Fig. 3, but for early winter (**a**) and late winter (**b**). Dots indicate statistical significance at the 5% level. Unit: number of unique ARs.

ARs per week in late winter (Fig. 5b). The impacts of ENSO are strong over the coastline during late winter and pronounced inland during early winter. ENSO's positive phase brings ~0.1 more unique ARs per week to California and Nevada in late winter. Positive RMM1 phases increase unique ARs in the Northwestern U.S. by ~0.1–0.2 per week compared to climatology, stronger in late winter, while the influence regions of RMM1 are not much different between early winter and late winter. RMM2 impacts are weak in the Western U.S. The impacts of PDO in the early and late winter are also similar to Fig. 2 middle and right. The PDO's positive phase leads to 0.05-0.1 more ARs per week in late winter over California, compared to climatology, with the opposite effect in the

negative phase (Fig. 5b). Also, the main difference in the influence of the PNA during early vs. late winter is similar to Fig. 2 middle and right, showing positive values (0.1–0.2 more unique ARs per week) extending further south into the equatorial Pacific in late winter compared to early winter, when the PNA is in positive phase.

In considering the four combined MJO phases, Fig. 6a, b illustrates a distinct contrast between early and late winter. It presents anomalies from composite analyses in the NDJ and JFM seasons (same method as Fig. 4), depicting the percentage change of AR temporal clustering compared to climatology. AR clustering during MJO phases 1 and 8 shows opposite patterns in early vs. late winter, especially in Washington/Oregon and

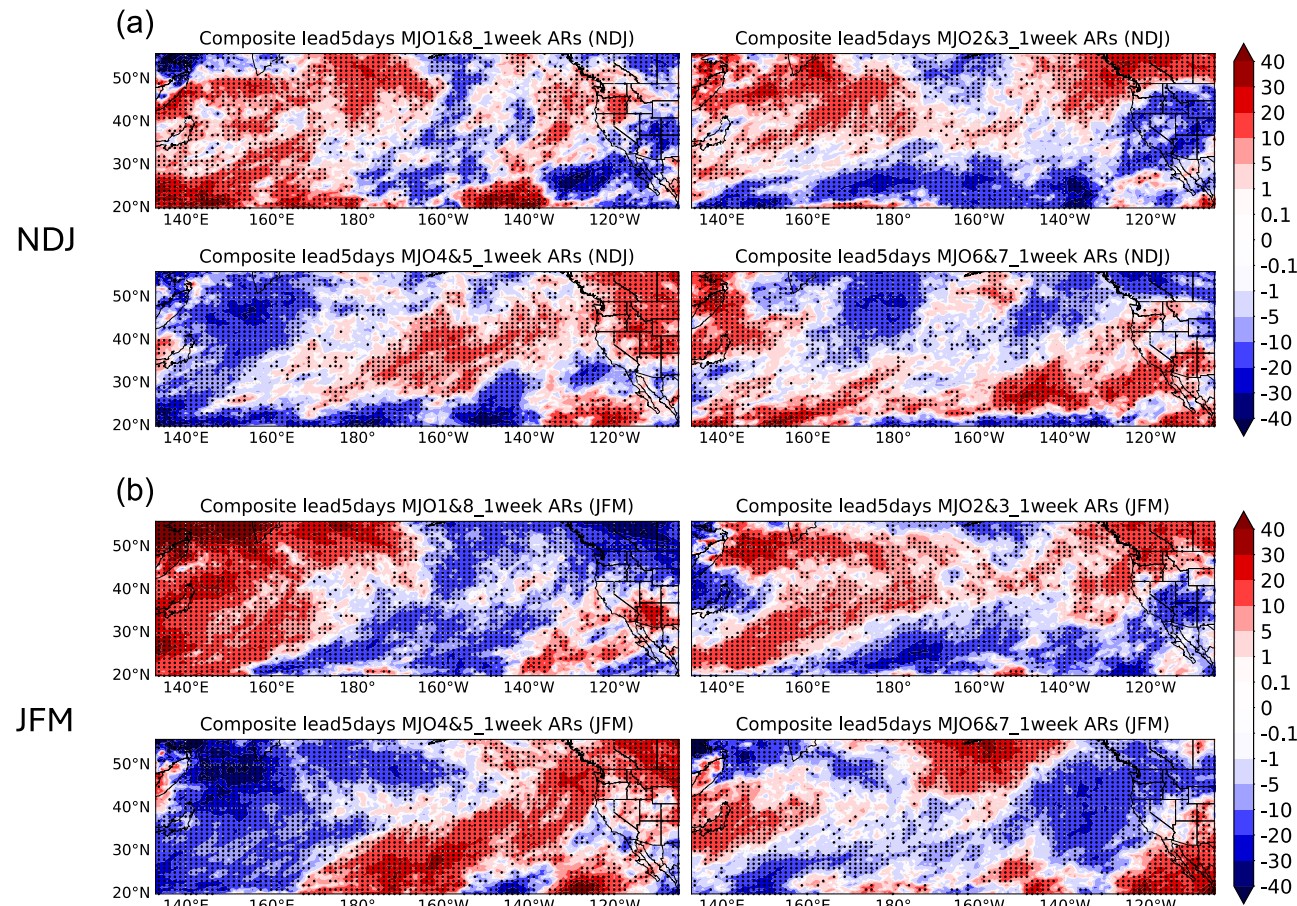

**Fig. 6 | Composite analysis of anomalies in the number of unique ARs based on four combined MJO phases during early and late winter.** Composite analysis in anomalies of the number of unique ARs in a 1-week window for early winter (**a**) and late winter (**b**) based on four combined MJO phases using the MERRA-2 AR dataset from 1982/1983 to 2020/2021. Relative difference compared to climatology (Unit: %). The dots represent statistical significance at the 5% level.

Arizona. In Arizona, late winter sees a 30% increase in AR clustering, while early winter sees a 20–30% decrease (Fig. 6). MJO phases 4 and 5 lead to a 10–20% decrease in temporal AR clustering in late winter over Southern California, but a 30% increase in early winter over the interior Western U.S. Over California, MJO phases 2 and 3 produce a north-south dipole in late winter and widespread negative anomalies in early winter. MJO phases 6 and 7 show a 30% increase in temporal AR clustering in early winter over Southern California and a 20% decrease in late winter over Northern California, Washington, and Oregon. The seasonality of MJO impacts is mainly from the different amplitudes and locations of both the MJO and basic states such as the westerly jet[35].

To see the differences more intuitively, Figs. 7 and 8 show the difference in the composite analysis of the number of unique ARs in the 1-week window between NDJ and JFM. In California, during negative phases of AO, QBO, ENSO, PDO, or PNA, the number of unique ARs is higher in early winter than in late winter, with significant effects seen for QBO and PNA (Fig. 7). For the combined MJO phases (Fig. 8), early winter shows around 0.3 more ARs per week compared to late winter in Washington/Oregon during phases 1 and 8, and phases 6 and 7 also show higher AR clustering over California during early winter. Therefore, by identifying significant differences in unique AR patterns between early and late winter, recognizing the impact of distinct climate modes phases, and adjusting forecast models, we will further enhance the accuracy of ARs S2S forecasts.

## AR IVT orientation
We now shift our focus from temporal clustering of unique ARs to AR IVT orientation of all AR days (mean IVT orientation; see Methods), of unique ARs (mean IVT orientation throughout their life cycle; see Methods), and of temporally clustered unique ARs (mean IVT orientation throughout their life cycle). In particular, we investigate the changes in AR orientation during the transition from early winter to late winter. Figure 9a shows a schematic of terms used to describe all AR days (black squares), unique ARs (light blue), and temporally clustered unique ARs (dark blue). Figure 9b shows the life-cycle IVT orientation of temporally clustered unique ARs (top row, left and middle) and all unique ARs (second row, left and middle) during NDJ and JFM. The right-side panels of Fig. 9b display the NDJ minus JFM differences, while the bottom panels highlight the first-minus-middle rows difference of Fig. 9b. We show AR IVT orientation climatology of all AR days in Supplementary Fig. 2. AR orientation is defined by the direction of mean IVT, measured clockwise from North (e.g., 0° indicates South to North, 90° indicates West to East; Fig. 9c). Therefore, in Fig. 9b right and bottom panels, a positive anomaly means a more westerly AR IVT, while a negative anomaly indicates a more southerly IVT. In general, the orientation of all ARs days increases from the Northwestern U.S. (~58°–59°, south-southwesterly) to Southern California/Arizona (~67°–68°, west-southwesterly), with a more noticeable shift in late winter (Supplementary Fig. 2). This aligns with Slinskey et al.[22]. The AR orientation difference between NDJ and JFM indicates that the IVT orientation of all AR days is significantly more westerly during early winter than late winter over much of the Western U.S. by around 2°–6° (Supplementary Fig. 2). This shift could be attributed to the seasonal variation in horizontal moisture transport from the subtropics to the extratropics[46,47]. Late winter (especially January and February) is often characterized by strong southerly winds due to the southernmost locations of the jet stream, leading to a more southerly orientation of ARs[48]. Understanding these differences is crucial for accurately forecasting the hydrological impacts of ARs across different times of the winter and this provides new insights into

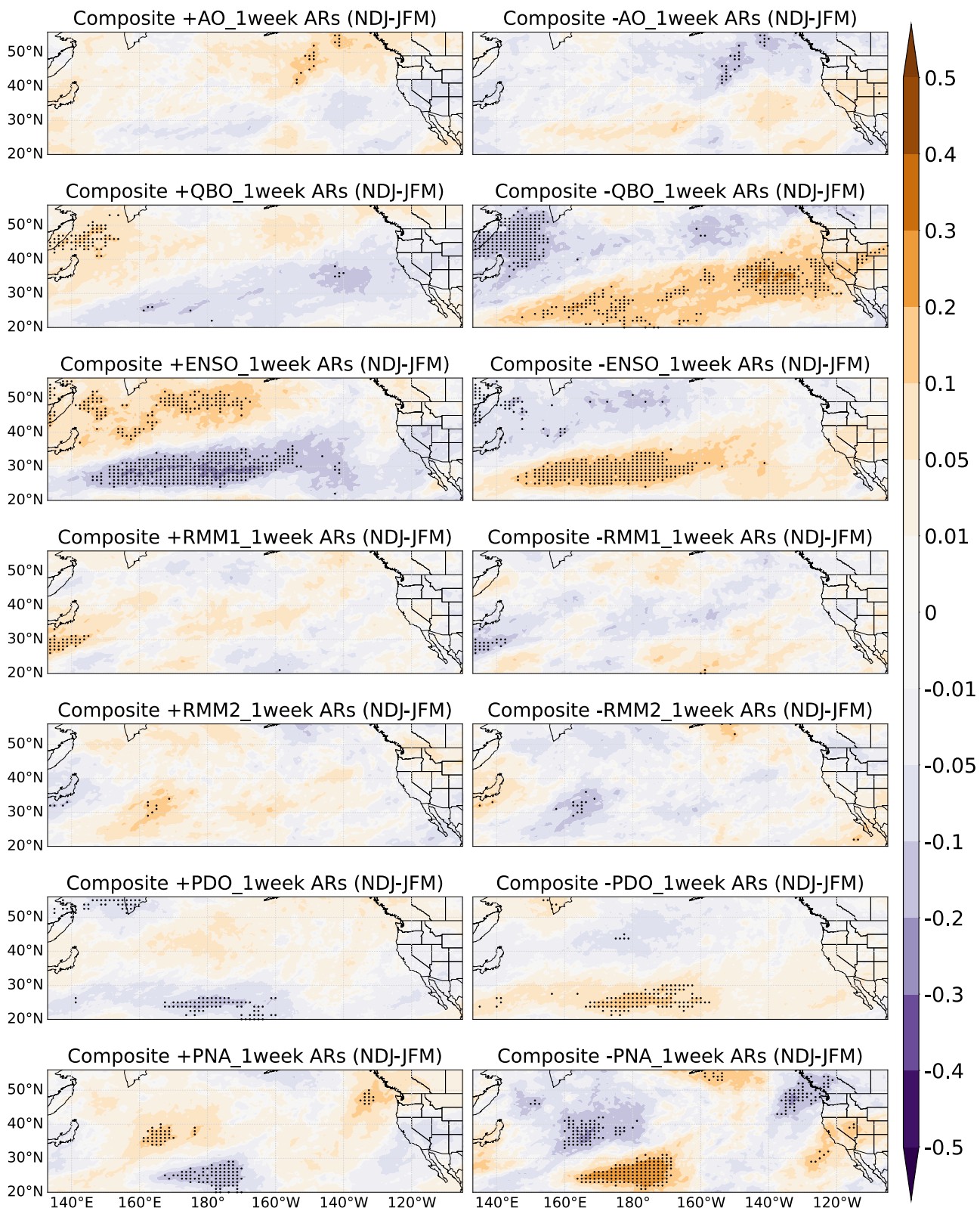

**Fig. 7 | Difference of composite analysis of anomalies in the number of unique ARs based on six climate modes between NDJ and JFM.** Composite analysis in anomalies of the number of unique ARs in 1-week window, but for early winter (NDJ) minus late winter (JFM), i.e., Fig. 5a minus Fig. 5b. Unit: number of unique ARs. The dots represent statistical significance at the 5% level.

AR orientation seasonal disaggregation in the western U.S. Based on the study of Picard and Mass[49], precipitation amounts over drainages in the Pacific Northwest can vary substantially with ~10° changes in the direction of AR-related wind flow influenced by the interaction with surrounding orography.

Consequently, the difference in AR orientation between NDJ and JFM can result in significant variations in precipitation amounts over the western U.S.

We now characterize the orientation of unique ARs (light blue squares in Fig. 9a; see Methods) and temporally clustered unique ARs (dark blue

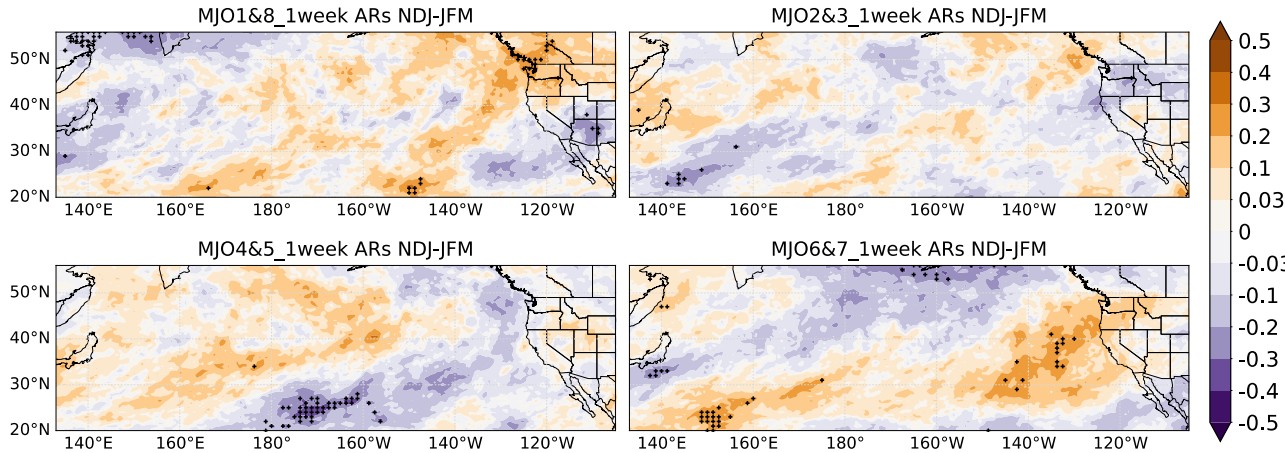

**Fig. 8 | Difference of composite analysis of anomalies in the number of unique ARs based on four combined MJO phases between NDJ and JFM.** Composite analysis in anomalies of the number of unique ARs in a 1-week window, but for early winter (NDJ) minus late winter (JFM), i.e., Fig. 6a minus Fig. 6b. Unit: number of unique ARs. The dots represent statistical significance at the 5% level.

squares in Fig. 9a). Their orientation also shows an increase in angle from the Northwestern U.S. (south-southwesterly) to Southern California/Arizona (west-southwesterly), similar to all AR days. In early winter, the temporal clustered unique ARs show a more southerly orientation along the North Pacific coast, while inland regions like Arizona and Southern California exhibit a more westerly orientation, though this is not statistically significant. Other inland regions show some sensitivity based on fixed windows (see Methods). The seasonality of the orientation of temporally clustered unique ARs is significantly different from the seasonality of the orientation for all AR days (Fig. 9b and Supplementary Fig. 2). Also, comparing unique ARs and clustered unique ARs, we can see temporal clustered unique ARs are more westerly than unique ARs, especially near the coast (Fig. 9b bottom panels).

Next, we investigate the influence of climate modes on the modulation of the life-cycle orientation of temporally clustered unique ARs. This investigation conducts a composite analysis that examines the life-cycle orientation of such events based on positive or negative phases of climate modes. We analyze early and late winter data (Fig. 10), and the anomalies show the composite analysis of temporally clustered AR orientation during positive and negative phases of climate modes, relative to climatology. A positive anomaly means a larger angle clockwise, so AR IVT is more westerly than climatology. The spatial patterns of the life-cycle orientation of temporally clustered unique ARs are distinctly influenced by the variability of AO, QBO, ENSO, MJO, PDO, and PNA. Specifically, AO shows strong seasonality of life-cycle orientation of temporally clustered unique ARs, with significant impacts in Central and Southern California in early winter and Northern California in late winter. Positive AO phases in early winter modulate to a strong westerly AR orientation over the Western U.S., especially over Central California by around 6°–8°, which can be attributed to AO being stronger in the early winter and then gradually weaken[50]. The positive AO is typically associated with stronger polar vortex and leads to stronger westerly winds across the mid-latitudes during this period. QBO negative phases cause a more southerly AR orientation in late winter. Notably, ENSO shows a strong modulation of AR orientation, which differs from its weak impacts on temporal AR clustering across the Western U.S. In early winter, the impacts of ENSO are mainly located in the Southwestern U.S. but not statistically significant, while in late winter, the significant impacts are mainly located on the Northern California and Pacific Northwest coast. MJO-RMM1 affects the Southwestern U.S. with more southerly (westerly) during the positive (negative) phase by ~6°–8° in early winter, while in the late winter, the main impacts are located on Pacific Northwest coast. MJO-RMM2 shows significant modulation over the Southwestern U.S. in late winter. In early winter, the positive PDO phase shows a more westerly AR orientation in Southern California and a more southerly

orientation in Nevada, though the seasonal difference is not statistically significant (Supplementary Fig. 3). The reverse is true for the negative PDO phase. Under PNA modulation, the life-cycle orientation of temporally clustered unique ARs shows strong seasonality over the interior Western U.S. In early winter, PNA impacts are centered on Arizona, while in late winter, they shift to the Great Basin. As for the seasonality, during the PNA positive phase, the orientation of temporally clustered unique ARs is significantly more westerly in Arizona (6°–8°) and the Great Basin (2°–6°) and more southerly over coastal Central California by around 2° during early winter. On the contrary, the situation is reversed during the negative phase of the PNA. Therefore, climate modes strongly influence the IVT life-cycle orientation of temporally clustered unique ARs, which in turn affects whether they enhance or reduce precipitation.

## Discussion and conclusions

We examined the temporal clustering of unique ARs over the North Pacific and Western U.S. region during the extended boreal winter period (NDJFM), specifically the modulation of temporal AR clustering and AR orientation by different climate modes of variability, and related differences during early (NDJ) and late (JFM) winter. Our analysis suggests that:

(1) Unique ARs exhibit temporal clustering in the Western U.S. during the extended boreal winter period, and this temporal clustering is conditioned on the occurrence of the climate modes.

(2) The impact of AO, QBO, ENSO, MJO, PDO, and PNA on clustering varies spatially. Notable shifts occur from early to late winter: the impacts of QBO on the Southwestern U.S. and PDO on California are mainly from the late winter, and for the AO, most of the impacts are in the early winter over Northern California.

(3) Regarding the MJO, four MJO combined phases show significant modulation of temporal AR clustering over the North Pacific and Western U.S. region. During NDJFM, MJO phases 1/8 and 4/5 most strongly modulate AR clustering and show opposite anomaly patterns in the Western U.S. MJO phases 4/5 corresponds to a significantly strong increase in temporal AR clustering events in the Northwestern U.S., including Washington/Oregon and Northern California. Also, the influence of MJO phases 1/8 and 6/7 on the temporal AR clustering in early winter is significantly different from that in late winter. This pattern is interesting since MJO significantly modulates AR clustering with the entire Western U.S., which differs from other works that have not observed such a widespread influence.

(4) IVT orientation of all AR days is significantly more westerly in early winter than in late winter over the Western U.S., indicating a strong seasonality of AR orientation. In contrast, the temporally clustered ARs

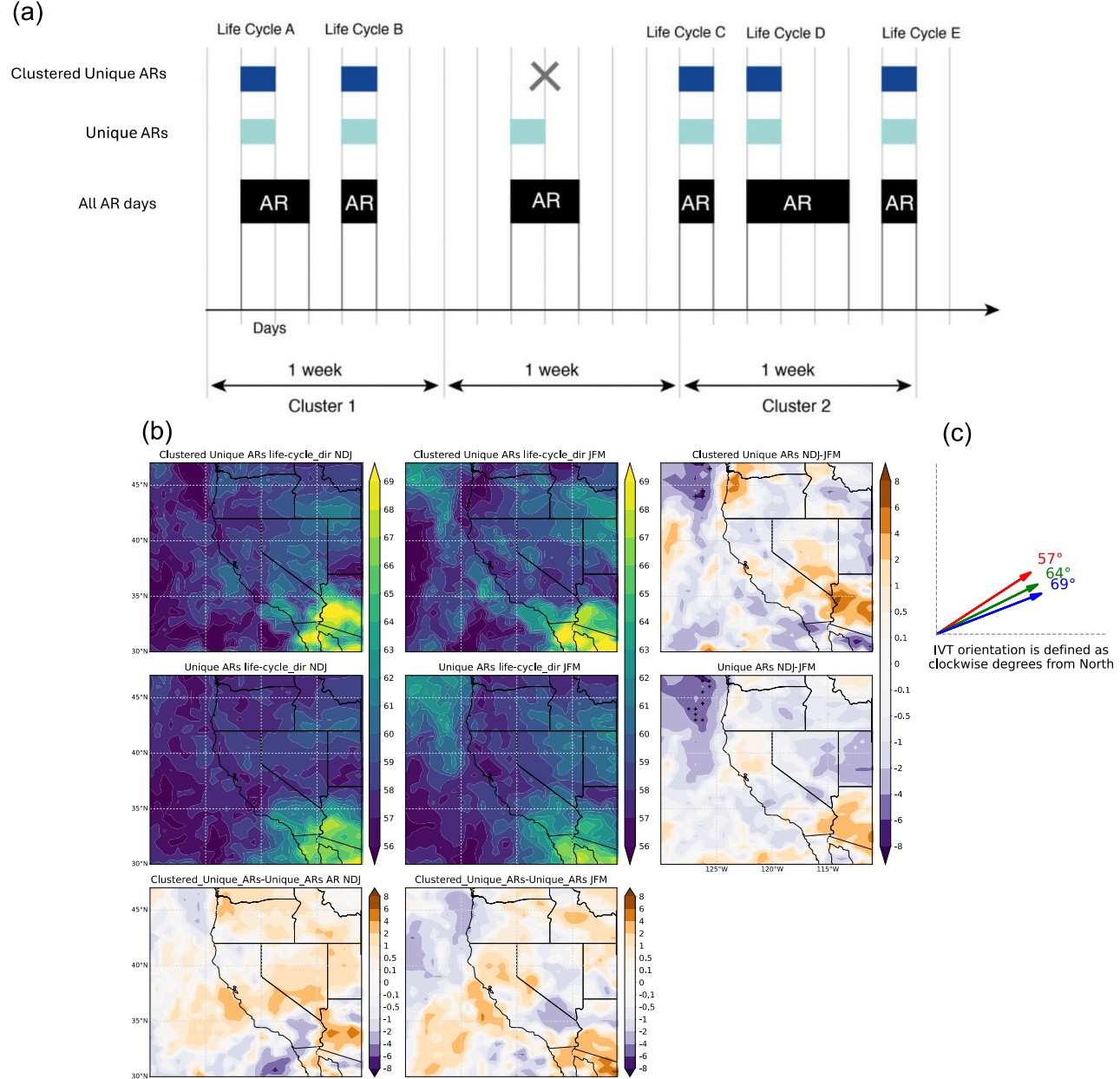

**Fig. 9 | Schematic overview of all AR days, unique ARs, and temporally clustered unique ARs, as well as IVT orientation across early and late winter. a** Schematic of terms used to describe all AR days, unique ARs, and temporally clustered unique ARs. The temporally clustered unique ARs satisfy the situation where there is more than one unique AR in a week window. **b** Life-cycle IVT orientation of temporally clustered unique ARs based on MERRA-2 AR dataset from 1982/1983 to 2020/2021 in early winter (NDJ, first row left), late winter (JFM, first row middle), and their difference (first row right). Life-cycle IVT orientation of unique ARs in early winter (NDJ, second row left), late winter (JFM, second row middle), and their difference (second row right). Life-cycle IVT orientation of temporally clustered unique ARs minus unique ARs in early winter (NDJ, third row left). Life-cycle IVT orientation of temporally clustered unique ARs minus unique ARs in late winter (JFM, third row right). Unit: degree. **c** IVT orientation diagram.

tend to exhibit a more southerly life-cycle orientation along the North Pacific coastline in early winter compared to late winter, whereas some inland regions experience a more westerly orientation in early winter. This pattern is interesting; since westerly winds are more conducive to precipitation in the Northwestern U.S., while southerly winds favor precipitation in the Southwestern U.S., temporally clustered ARs in early (late) winter are more likely to lead to the most significant hydrologic impacts in the Northwest (Southwest).

(5) ENSO shows a strong modulation of the IVT life-cycle orientation of temporally clustered unique ARs, which differs from its weak impacts on temporal AR clustering, which is supportive of other work, such as Guirguis et al.[23].

These results yield a new understanding of temporal AR clustering over the North Pacific and Western U.S. region, including the seasonality of clustering and its relationship to various climate modes of variability. In this study, results from both the Cox regression analysis and composite analysis are similar, and both perspectives help characterize temporal clustering of unique ARs, their seasonality, and their relationship to climate mode variability during the extended boreal winter season over the North Pacific and Western U.S. region. These results are also consistent with study of Zhou et al.[13], which employs machine learning techniques to define the temporal AR clusters and identifies a PNA-like pattern exerting significant influence.

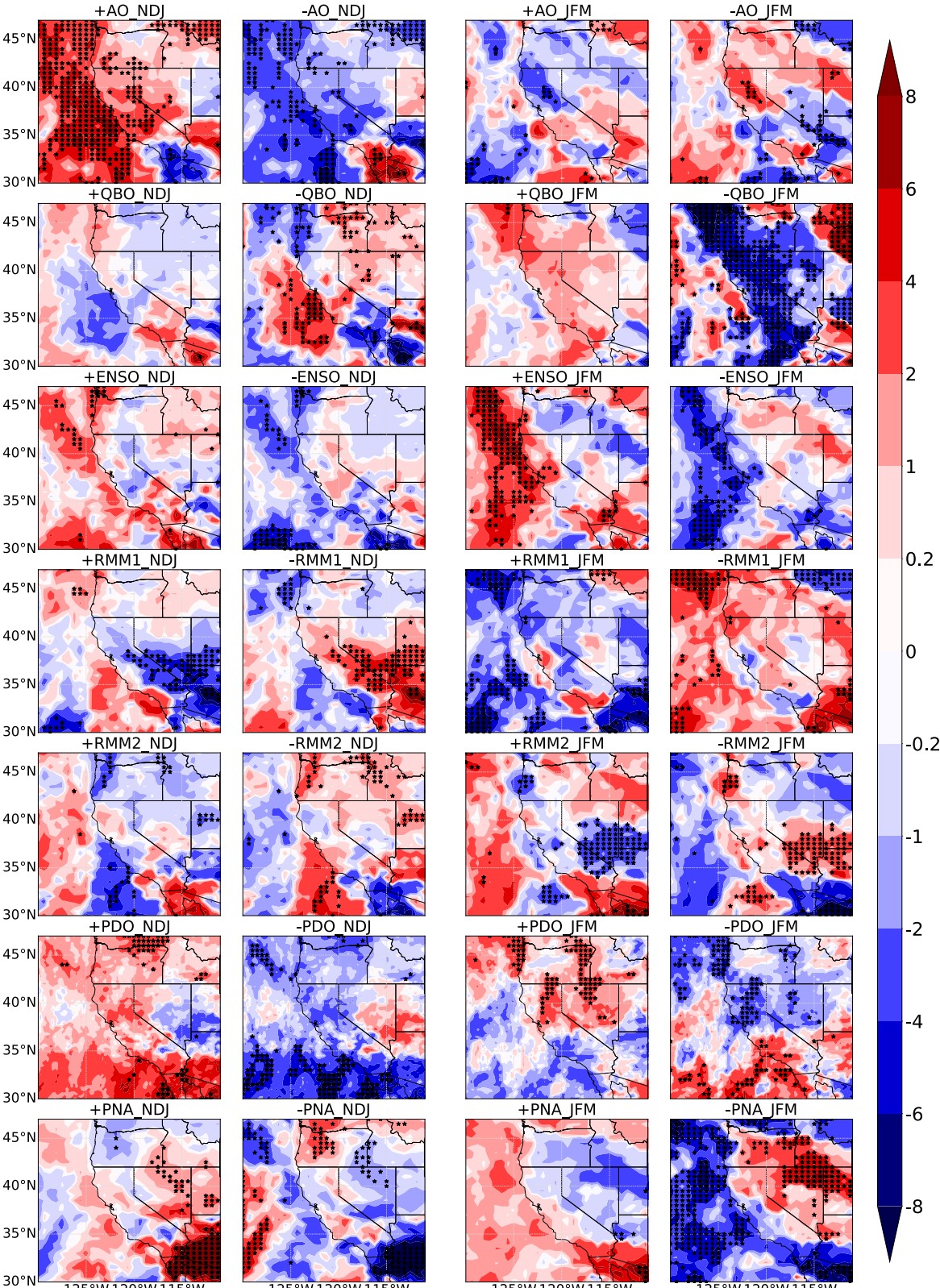

**Fig. 10 | Climate modes modulate IVT orientation of temporally clustered unique ARs.** Composite analysis in anomalies of the life-cycle IVT orientation of temporally clustered unique ARs during early winter (NDJ) and late winter (JFM). The analysis based on positive (columns 1,3) and negative phases (columns 2,4) of AO, QBO, ENSO, MJO-RMM1, MJO-RMM2, PDO, PNA (removed ENSO signal), using the ERA-5 AR dataset from 1940/1941 to 2017/2018 (for PDO analysis) and MERRA-2 AR dataset from 1982/1983 to 2020/2021 (for other climate modes analysis). The anomalies represent a composite analysis of temporally clustered AR orientation based on the positive and negative phases of climate modes, minus the climatology of temporally clustered AR orientation. Dots indicate statistical significance at the 5% level. Unit: degree.

This study also emphasized the orientation of ARs, which will influence the spatial distribution of precipitation in a region. The strong seasonality of AR orientations between early and late winter can result in shifts in the areas that receive the heaviest precipitation, and the topography of the region can interact with the orientation of ARs to influence local weather patterns[23,25,26]. Changes in orientation may affect how they interact with geographical features, and orientation at landfall can determine whether and how an AR penetrates inland.

Furthermore, the modulation of life-cycle IVT orientation of temporally clustered unique ARs by different climate modes provides us with ideas for forecasting AR-related precipitation. Specifically, since the impacts of ARs and the related precipitation are closely related to their orientation, climate modes that do not strongly influence temporal AR clustering or frequency can strongly modulate AR precipitation through changes in IVT orientation. For instance, ENSO does not show the influence of AR temporal clustering in the Pacific Northwest but significantly makes the life-cycle IVT orientation of temporally clustered ARs more west-southwesterly during the El Nino phase and more south-southwesterly during the La Nina phase during early winter. Therefore, it will be equally important to consider the impact of AR orientation and temporal AR clustering, frequency, and magnitude in future research. This comprehensive analysis will enhance our understanding of AR impacts under climate modes' modulation with the goal of informing future improvements in precipitation forecasts.

## Methods
### Climate modes
To capture the impacts of climate mode variability on ARs over the Western U.S., we consider six dominant climate modes at the daily scale, including the AO, the QBO, the ENSO, the MJO, the PDO, and the PNA. Each of these climate modes significantly modulates precipitation and AR activity in the Western U.S.[23,51]. We consider these climate modes dominant based on the literature review of their significant impacts on atmospheric rivers and precipitation over the western U.S. and on their utilization in the S2S operational forecasts. For instance, Guan and Waliser[41] found that the AO, ENSO, MJO, and PNA significantly impact AR frequency and precipitation in the western U.S. Specifically, the increased activity over California was linked to the negative phase of both the PNA and AO. ENSO modulates AR frequency along the western U.S. coast. The AR activity in the California Sierra Nevada is shown to be significantly strengthened when MJO convection is in phase 6[52]. The PNA-like pattern strongly correlates with temporal AR clustering in the western U.S.[13]. In the operational forecast, MJO and QBO strongly modulate subseasonal AR activity and precipitation in California, with reduced AR activity during easterly QBO and increased activity during westerly QBO in the mid-to-late winter[36]. Additionally, Gershunov et al.[43] reported that AR activity along the west coast of North America generally increases during positive phases of the PDO and decreases during negative phases. Furthermore, we consider two sets of AR datasets separately for long-period cycle (i.e., PDO) and short-period cycle climate modes.

Due to the constraints of the data timeframe and facilitate comparative research utilizing the same AR dataset (MERRA-2 based AR catalog)[53], the analysis based on AO, ENSO, MJO, QBO, PNA and is conducted for the period of water years 1983–2021 and MERRA-2 based AR catalog (see Identify unique ARs on a daily scale), while the analysis based on PDO is period of water years 1941–2018 and ERA5 based AR catalog (see Identify unique ARs on a daily scale).

The daily AO and PNA indices are downloaded from the NOAA Climate Prediction Center (CPC), covering January 1950 to the present (https://www.cpc.ncep.noaa.gov/; or https://ftp.cpc.ncep.noaa.gov/cwlinks/). Similar examples of data usage include[54]. For ENSO, the daily NINO3.4 index is calculated from daily NOAA OI SST V2 High Resolution Dataset anomalies from 1981 to the present (https://www.psl.noaa.gov/data/gridded/data.noaa.oisst.v2.highres.html[55]), defined by average SST anomalies over 5°N to 5°S, 170°W to 120°W[56]. We use linear regression to remove the influence of the ENSO signal on the PNA, since PNA is strongly related to phases of ENSO, projection highly depends on ENSO. For instance,

during El Niño events, the PNA tends to exhibit a positive phase, while during La Niña events, it tends to exhibit a negative phase[57,58]. The daily QBO index is calculated from daily-averaged Modern-Era Retrospective analysis for Research and Applications, Version 2 (MERRA-2) 50-hPa zonal mean zonal wind anomalies[59] over the tropics (10°S–10°N), covering 1980 to the present[60]. The daily MJO index is obtained from the Australian Bureau of Meteorology (http://www.bom.gov.au/climate/mjo/), including RMM1, RMM2, phase, and amplitude, which covers June 1974 to the present. The real-time RMM1 and RMM2 are defined by empirical orthogonal functions (EOFs) of the combined fields of satellite-observed outgoing longwave radiation (OLR), near-equatorially averaged 850hPa zonal wind (u850), and 200-hPa zonal wind (u200) data[42]. RMM1 and RMM2 jointly indicate both the phase and amplitude of the MJO. Individually, RMM1 indicates intensified convection at the Maritime Continent longitudes, while RMM2 signifies heightened convection over the Pacific Ocean[42]. To find the largest time overlap of the above climate modes and the MERRA-2 AR events dataset, we use the period of water years 1983-2021 extended winter from November to March, which is the season with the highest climatological frequency of AR events in the Western U.S.

In order to investigate the modulation effect of the low-frequency PDO climate mode, we consider a longer-period monthly PDO index from University of Washington, covering 1900 to 2018 (http://research.jisao.washington.edu/pdo/PDO.latest). This PDO index is derived from the UKMO Historical SST dataset for 1900–1981, Reynold's Optimally Interpolated (OI) SST (V1) for 1982 to 2001, and OI SST Version 2 (V2) from 2002 to 2018. Then, we use linear interpolation to obtain the daily PDO index[23]. To find the largest time overlap between the PDO index and the longest period of the AR events dataset based on ECMWF Reanalysis v5 (ERA5), we use the period of water years 1941-2018 winter from November to March.

### Identify unique ARs on a daily scale
To detect AR events over the Western U.S. and Pacific Ocean, we employ the Guan and Waliser[41,61,62] algorithm. This algorithm stands out as one of the few globally accessible methods, defining AR based on IVT intensity, direction, and geometry. This AR identification method also shows strong agreement with independently developed methods found in other studies[63] as supported by results from the Atmospheric River Tracking Method Intercomparison Project (ARTMIP)[53].

To better study the impact of long-period cycle (i.e., PDO) and short-period cycle climate modes and make full use of existing data, we consider two sets of AR datasets separately. One is the commonly used MERRA-2-based AR catalog downloaded from Global Atmospheric Rivers Dataverse (https://dataverse.ucla.edu/dataverse/ar), which also provides AR track IDs to identify unique AR events. This AR dataset covers from 1980 to 2021, with 6-hourly global data with a resolution of 0.5° x 0.625°. We use this MERRA-2-based AR dataset to capture the impacts of short-period climate modes of variability on unique ARs (i.e., AO, QBO, ENSO, MJO, PNA ~ MERRA-2 AR) and enable comparative research using the same AR dataset[53]. Another is the newly published long-period ERA5-based AR catalog downloaded from the same website (https://dataverse.ucla.edu/dataverse/ar). This AR dataset goes from 1940 to 2022, with 6-hourly global data and a resolution of 0.25° x 0.25°. We utilize the ERA5-based dataset to capture the impacts of climate models characterized by interdecadal variability, such as the PDO, on unique ARs (i.e., PDO ~ ERA5 AR). To identify AR events at daily scale, we consider in each grid cell that if there is an AR object at any timestep (4 timesteps per day) during a day, then the day is considered an AR day, which is used for AR orientation analysis. We identify time series of unique ARs that contain occurrence or non-occurrence (represented as '1' or '0') at a daily scale in each grid (light blue squares in Fig. 9a).

### Cox regression analysis
To identify the presence and magnitude of unique AR temporal clusters, we applied Cox regression analysis[64,65], which has been used to identify temporal

clustering in hydroclimatological studies, heavy precipitation, and floods in the Potomac River, the Central U.S., and Europe[15–18,20,65–67]. The Cox process is a generalization of a Poisson process. Specifically, a Poisson process is a memoryless process, while the Cox process belongs to the family of clustered processes, indicating the occurrence of one event has connections with the subsequent one. In a Cox process, the rate of occurrence parameter $\lambda$ is not deterministic but a random function[16]. The rate of occurrence $\lambda$ for the $t^{th}$ day ($t = 1,2,\dots$, 151 for NDJFM; $t = 1,2,\dots$, 92 for NDJ; $t = 1,2,\dots$, 90 for JFM) in the $i^{th}$ year can be described by the following equation:

$$\lambda_i(t) = \lambda_0(t) \exp\left[\sum_{q=1}^{m} \beta_q Z_{iq}(t)\right]$$

where $\lambda_0$ is the baseline hazard, which is a non-negative function for time. $Z_{iq}$ are time-varying $q^{th}$ covariates, i.e., AO, QBO, ENSO, MJO, PNA, and PDO; $m$ is the number of covariates. $\beta_q$ indicates the regression coefficient of each covariate ($q = 1,\dots,m$). Since the Cox process is the semiparametric model, where the baseline hazard function is non-parametric and the exponential part is parametric, we calculate $\beta$ coefficients based on the maximum partial likelihood method[16]. We use $\beta_q$ to quantitatively measure the influence of climate modes on the cluster distribution of unique ARs. The climate modes of variability $Z_{iq}$, which alternates between positive and negative phases, dictates the active and quiet periods of unique ARs. Similar to the methodology employed by Mallakpour et al.[16], we calculate a seven-day moving average for each of the daily climate indices and utilize them as predictors in our regression model.

In this study, to consider the most significant covariates in each grid, we cover all possible combinations (i.e., 32 combinations in total, calculated by permutations and combinations) of AO, QBO, ENSO, MJO, and PNA. We also took into account the correlation among the different predictors in the context of collinearity. Anderson and Burnham[68] recommended using a cutoff value of |0.95| for excluding predictors, while retaining those with seemingly significant correlations. In this study, the correlation values among these AO, PNA, MJO, QBO, ENSO are generally much smaller than |0.3| (Supplementary Table 1). Therefore, we do not believe that collinearity among the climate indices has impacted our results. Then, to find the best Cox regression model, we select a regression model with the smallest Akaike information criterion (AIC[69]).

Regarding PDO, we compute only based on the PDO index and ERA5 AR dataset. Noted that since we use two separate Cox regression analysis for AO, QBO, ENSO, MJO, PNA and PDO, the coefficients from the first regression (i.e. AO, QBO, ENSO, MJO, PNA) cannot directly be comparable to those from the second (i.e., PDO). All calculations were performed using the survival package in R[70].

In the analysis of modulation of temporal AR clustering by several climate modes in Fig. 2, the extent of this modulation is quantified using the Cox regression coefficient $\beta$ for AO, QBO, ENSO (considering NINO3.4 here and following), MJO, PDO, and PNA (excluding ENSO signal here and following). We define the positive or negative phases of each mode based on whether the indexes of climate modes are positive or negative. A positive $\beta$ indicates that when a given climate mode is in a positive phase, the rate of occurrence of unique ARs will be high, suggesting an active period of more unique ARs. On the contrary, a negative $\beta$ indicates that when climate modes are in a negative phase, the rate of occurrence of more unique ARs will be high, suggesting the active period of more unique ARs[16,20]. In the Cox regression analysis, we do not set a window to identify temporal clustering. Instead, the significant $\beta$ coefficient indicates the presence of temporal clustering among unique ARs, independent of any window setting.

## Setting number of unique ARs anomalies in a 1-week window
We use a 1-week window to measure the temporal compounding of ARs in composite analysis, which is to establish a foundational framework for future research on subseasonal AR clustering and prediction within a dynamic statistical forecasting framework. Inspired by the clustering

framework presented by Fish et al.[10,11], we implement a 1-week window to identify AR families here. While Fish et al.[10,11] classify AR events as an AR family if two or more AR objects occur within the 5-day aggregation period, we adapted similar concepts to define AR families. Therefore, 1-week is an appropriate window to identify clustering. While a longer window could offer additional insights, our approach provides a starting point for understanding temporal compounding on a manageable scale.

The underlying null hypothesis at each grid cell in Figs. 4 and 5 is that there is no significant difference in the number of unique ARs in a 1-week window during the winter season (NDJFM in Fig. 4, NDJ or JFM in Fig. 5) between the composite analysis based on positive or negative phases of climate modes (or combined MJO phases in Fig. 4) and the long-term average from 1982/1983 to 2020/2021. In other words, it posits that any observed anomalies in the number of unique ARs at each grid cell are due to random variability rather than an effect of the climate modes phases. Statistical significance at the 5% level indicates that there is less than a 5% probability that the observed anomalies are due to random chance.

## AR orientation of all AR days and of temporally clustered unique ARs
We consider AR orientation of all AR days based on mean IVT orientation. In this case, each AR day gets an equal weight when averaging. This approach also facilitates comparisons with other studies. For example, the mean orientation of AR days tends to be more southerly in the northwestern U.S. and more westerly in the southwestern U.S. Additionally, this metric is widely used in climatological studies and operational forecasting, as it helps identify prolonged periods of moisture transport and precipitation.

Note that we characterize the orientation of unique ARs and of temporally clustered unique ARs by considering the mean IVT orientation throughout their life cycle. A long-lasting AR would get the same weight as a short-lasting AR. When there is more than one unique AR occurring within fixed one-week windows, we classify them as temporal clustered unique ARs, depicted as dark blue squares in Fig. 9a. Therefore, in Fig. 9a, the 3rd AR (black squares) is not considered part of a cluster. The climatology of the life-cycle IVT orientation of temporally clustered unique ARs is calculated based on averaging all ARs belonging to clusters. Our results are not sensitive to the start date of the one-week interval (Supplementary Fig. 4).

## Statistical significance test
To evaluate the statistical significance of composite analysis patterns associated with each climate mode, we employ bootstrapping. This involves generating distributions of multiple random composite analysis patterns (e.g., 1000 iterations), each derived from an equal number of positive and negative phases of the climate modes. By assessing the extent to which the composite analysis patterns deviate from the distribution tails, we determine the statistical significance of the results.

To test the statistical significance difference between composite analysis patterns in the early and the late winter (NDJ vs. JFM), we use the Student's t-test.

## Data availability
The daily NINO3.4 is calculated from daily NOAA OI SST V2 High Resolution Dataset anomalies (https://www.psl.noaa.gov/data/gridded/data.noaa.oisst.v2.highres.html). The daily AO and PNA are downloaded from the NOAA Climate Prediction Center (CPC) (https://www.cpc.ncep.noaa.gov/products/precip/CWlink/pna/pna.shtml; https://ftp.cpc.ncep.noaa.gov/cwlinks/). The daily MJO is from the Australian Bureau of Meteorology (http://www.bom.gov.au/climate/mjo/). Monthly PDO index from the University of Washington (http://research.jisao.washington.edu/pdo/PDO.latest). The daily QBO is calculated by MERRA-2 50hPa zonal mean zonal wind anomalies (https://disc.gsfc.nasa.gov/datasets?keywords=u&page=1). AR dataset based on MERRA-2 and ERA5 are from https://doi.org/10.25346/S6/YO15ON.

## Code availability

All Cox regression calculations were performed using the survival package in R (Theerneau 2014, https://cran.r-project.org/web/packages/survival/index.html). The analysis code utilized in this study can be obtained here https://github.com/CW3E/ARclustering or by reaching out to the corresponding author upon request.

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

## Acknowledgements
We thank editors Dr. Heike Langenberg, Dr. Alireza Bahadori, Dr. Kathleen Williamson, and two reviewers for their insightful comments. This research was supported by the California Department of Water Resources (CA DWR) Atmospheric River Program. Z.Y. thanks Dr. Ziyu Feng for insightful comments and suggestions on an earlier version of this study. A.S. and B.G. acknowledge support from National Aeronautics and Space Administration (NASA) Grant 80NSSC22K0926. The AR database (https://doi.org/10.25346/S6/YO15ON) was provided by Bin Guan via the Global Atmospheric Rivers Dataverse (https://dataverse.ucla.edu/dataverse/ar). Development of the AR database was supported by NASA and the CA DWR.

## Author contributions
Conception of the work: Z.Y. and M.J.D. B.G., M.J.D., and A.G. provided the dataset and analysis suggestions. Z.Y. collected, analyzed, and visualized the data and wrote the draft manuscript. M.J.D., A.S., J.W., C.M.C., A.G., K.G., E.S., B.G., L.D.M., and F.M.R. provided feedback and contributed to manuscript revisions. F.M.R. and L.D.M. acquired funding support.

## Competing interests
The authors declare no competing interests.
