## [Transparent Peer Review file · Communications Earth & Environment]

Seasonality and climate modes influence the temporal clustering of unique atmospheric rivers in the Western U.S.

Corresponding Author: Dr Zhiqi Yang

Version 0:

Decision Letter:

Dear Dr Yang,

Please allow me to apologise for the delay in sending a decision on your manuscript titled "Temporal clustering of unique atmospheric rivers impacting the Western U.S.: seasonality and climate modes modulation". It has now been seen by 2 reviewers, and we include their comments at the end of this message. They find your work of interest, but some important points are raised. We are interested in the possibility of publishing your study in Communications Earth & Environment, but would like to consider your responses to these concerns and assess a revised manuscript before we make a final decision on publication.

We therefore invite you to revise and resubmit your manuscript, along with a point-by-point response that takes into account the points raised. Specifically, please strengthen your discussion of the state-of-the-art, and substantially deepen your interpretation and conclusions, based on the reviewers' comments. We also encourage you to discuss the importance and implications of your work. Please highlight all changes in the manuscript text file.

Please submit your point-by-point responses as a separate file, distinct from your cover letter where you can add responses to the Editors' comments that you do not want to be made available to the reviewers. Word files are preferred.

Important: The response to reviewers must not include any figures, tables or graphs. If you wish to respond to the reviewer reports with additional data in one of these formats, please add them to the main article or Supplementary Information, and refer to them in the rebuttal. Due to current technical limitations, any figures, tables, or graphs embedded in your rebuttal will not be included in the peer review file, if published.

Please use the following link to submit your revised manuscript, point-by-point response to the referees' comments (which should be in a separate document to any cover letter), a tracked-changes version of the manuscript (as a PDF file) and the completed checklist:

Link Redacted

We hope to receive your revised paper within six weeks; please let us know if you aren't able to submit it within this time so that we can discuss how best to proceed. If we don't hear from you, and the revision process takes significantly longer, we may close your file. In this event, we will still be happy to reconsider your paper at a later date, as long as nothing similar has been accepted for publication at Communications Earth & Environment or published elsewhere in the meantime.

Please do not hesitate to contact us if you have any questions or would like to discuss these revisions further. We look forward to seeing the revised manuscript and thank you for the opportunity to review your work.

Best regards,

Heike Langenberg, PhD
Chief Editor
Communications Earth & Environment

EDITORIAL POLICIES AND FORMATTING

Editorial Policy: [Policy requirements](https://www.nature.com/documents/nr-editorial-policy-checklist.pdf) (Download the link to your computer as a PDF.)

- Behavioural and social science
- Ecological, evolutionary & environmental sciences
- Life sciences

<https://www.nature.com/documents/nr-reporting-summary.zip>

Furthermore, please align your manuscript with our format requirements, which are summarized on the following checklist: [Communications Earth & Environment formatting checklist](https://www.nature.com/documents/commsj-phys-style-formatting-checklist-article.pdf)

and also in our style and formatting guide [Communications Earth & Environment formatting guide](https://www.nature.com/documents/commsj-phys-style-formatting-guide-accept.pdf) .

*** DATA: Communications Earth & Environment endorses the principles of the Enabling FAIR data project (<http://www.copdess.org/enabling-fair-data-project/>). We ask authors to make the data that support their conclusions available in permanent, publically accessible data repositories. (Please contact the editor if you are unable to make your data available).

All Communications Earth & Environment manuscripts must include a section titled "Data Availability" at the end of the Methods section or main text (if no Methods). More information on this policy, is available at <http://www.nature.com/authors/policies/data/data-availability-statements-data-citations.pdf>.

If a community resource is unavailable, data can be submitted to generalist repositories such as [figshare](https://figshare.com/) or [Dryad Digital Repository](http://datadryad.org/). Please provide a unique identifier for the data (for example a DOI or a permanent URL) in the data availability statement, if possible. If the repository does not provide identifiers, we encourage authors to supply the search terms that will return the data. For data that have been obtained from publically available sources, please provide a URL and the specific data product name in the data availability statement. Data with a DOI should be further cited in the methods reference section.

REVIEWER COMMENTS:

Reviewer #1 (Remarks to the Author):

This paper looks at the effects of climate modes and early vs. late winter seasonality on both temporal clustering and IVT orientation in ARs affecting the North Pacific Ocean and Western US. While many of these factors have been examined separately in the AR literature, this is the first to look at interactions between them. The authors have a clear understanding of the existing literature, the analysis is strong, and overall this paper would be a positive contribution to the field.

My main issue with this paper is the complete lack of interpretation. In many cases the authors simply list differences or patterns in the figures without contextualization or comparison, and while the “what” and “how” were well described, I was often left missing the “why,” which left the paper feeling much more like a disjointed collection of facts than a cohesive narrative. I also have some concerns with the metrics of temporal compounding, as detailed in my comments below. I recommend minor revisions.

Major comments:

1. Throughout the paper, the work on AR orientation is really framed as an afterthought, not a main contribution. The abstract includes only one of the two central questions you pose in the introduction (and I don't like the repetition of “yet the enigma persists” in both the abstract and the body of the paper). It is not until the very last paragraph of the conclusion that you mention why you are examining both: i.e., both are major determinants of AR impacts and climate modes can modulate the two in different (and sometimes opposing) ways. AR orientation deserves a more complete treatment in the abstract, introduction, and interpretation of results.
2. I would encourage the authors to try to, in their own words, answer “This finding is important because...” for the discussion of every single figure. The stated goal of this paper is to improve seasonal-to-subseasonal forecasting; how do different composite patterns support that goal? What can we learn about anticipated impacts? What does it mean for forecast skill when modes have similar vs. contrasting anomaly patterns? There is a lot of information in these figures; your goal is not to summarize it all, but to extract the key insights and tell the reader what to care about.
3. The authors assume that the number of unique ARs in a 1-week window can be used as a measure of temporal compounding. I am not convinced that this is an appropriate or accurate metric. Many of the paper's main results (Figs 3-8) depend on this logical leap, so further explanation and support of this choice is required.
4. Fitting two Cox regressions, where one regression considers five modes and the other considers only PDO, means that the coefficients from the first regression are not directly comparable to those from the second. It is misleading to put them on the same plot with the same color scale without a note acknowledging this expected difference. It also means any comparative interpretation of the magnitude of the coefficients is flawed. Very little information is provided about the regressions, and more detail is needed about the regression forms that were tested for AO, PNA, MJO, QBO, and ENSO.

Minor comments:

5. Line 57: should be “AR families”, not family
6. Line 58-59: This seems circular; the presence of clustering suggests clustering? Also, it is awkward to define “independent ARs” at the same time as defining ARs within bursts.
7. Line 71: “temporally clustered ARs”, not AR
8. Line 74: Likely do not need all these citations; 19-20-21 seem very similar, as do 22-23-24-25-29. Also “floods on the Potomac River” is a bit jarring because it is so much more geographically specific than the others mentioned and it is a measure of impact rather than hazard. Something like “streamflow in the eastern US” would be more in line with the way you discuss the other references.
9. Line 80: Add a citation number for Slinsky et al.
10. Line 78-104: This could be set up better.
 - a. Please streamline the citations more. The paragraph feels very repetitive (“Paper A said this. Paper B said this. Paper C...”) without a clear thread. What should my takeaways be—the most common/most impactful orientation by global region? Or just the fact that impacts are influenced by orientation? If the goal is the former, please state the takeaways explicitly; if it is the latter, you could make the point without nearly as much distracting detail.
 - b. The entire introduction has focused on temporal compounding up to this point, with no previous mention of orientation, so the transition between this paragraph and the preceding one is a little abrupt.
 - c. The connection between [32] and orientation is unclear.
 - d. The description of [33] is a run-on sentence.
 - e. For [9], where is the Teifi catchment?
 - f. Line 101: This is really important! This is a critical connection that explains why an examination of orientation has value for forecasting, and it should be highlighted accordingly.
11. Line 105: I do not think you actually answer your two main motivating questions as posed, so it would be helpful to both you and the reader to (a) make them more specific and (b) explicitly state the value of answering each.
 - a. “Why do ARs cluster in time?” what is the interacting effect of climate modes and seasonality on AR clustering? (it is important to answer this question because...)
 - b. “What are the characteristics of the AR orientation associated with the temporally clustered ARs?” what is the interacting effect of climate modes and seasonality on AR orientation? (it is important to answer this question because...)
12. Line 114: two “and”s
13. Line 115-116: Incomplete clause. Should it be seasonal disparities “exist” between?
14. Line 118-125: It's not really the analysis of AR clustering that accomplishes this, it's the discovery of relationships between factors with high subseasonal predictability and AR clustering/orientation.
15. Line 134: The phrasing “by around” is confusing. I think you mean “to around”; otherwise it reads like the number of unique ARs along the coastline is $1.4 - 1 = 0.4$. Also they should all be in units of “ARs per week”, not “AR per week”
16. Fig 2: Column and row labels would be helpful.
17. Line 137-159: This is such a disjointed paragraph. Can you add interpretation—how does this compare to previous literature? Why do these specific spatial patterns matter?
 - a. Why is PNA signal so much stronger than the others?

b. Line 157: This is the topic sentence for this paragraph, it would help a lot to move to the top of the paragraph so the reader knows what to expect.

18. As a general comment, the (over)use of “specifically”, “furthermore”, “moreover”, etc. between the results of different climate modes is distracting. Transition words are not necessary between elements of a list.

19. Line 160: “the number of unique ARs”: are you counting AR days or AR events? If events, are you counting starts?

20. Line 175-176: This is a helpful global observation that belongs at the top of the paragraph, before all of the details associated with specific modes. Similar to above, please provide interpretation for each climate mode (“This pattern is interesting/informative/noteworthy because...”)

21. Line 178: Why, out of all the climate modes, does MJO deserve a closer examination?

22. Line 195: Please see comments about lack of interpretation for this entire section.

23. Line 196: “the modulation of unique ARs temporal clustering conditioned upon climate mode variability during boreal winter” is very awkward phrasing, please revise.

24. Why does Fig 4, and only Fig 4, have the absolute value composite plots in addition to the anomaly plots? What critical information does this provide?

25. Why does Fig 6 have units of % when Figs 3, 4, 5, 7, and 8 are all in units of unique ARs?

26. Figs 7-8: These are differences, not anomalies, so it would be nice to have a different color scale.

27. You are inconsistent in the way you describe orientation: sometimes using a meteorological convention (ex. “south-southwesterly”), sometimes using “from X to Y”, sometimes using geographic landmarks (ex. “angled towards the Pacific Northwest”), sometimes using degrees, etc. This is an issue throughout the results + in the introduction and makes it hard to draw useful and informative comparisons.

28. Line 274: How do your results compare to previous literature about AR orientation in the western US? Has anyone else done a seasonal disaggregation before? If not, it is worth highlighting that as a novel contribution.

29. Line 277-279: You are very close to telling me why these results matter! But you don’t finish the thought, which leaves this sentence feeling out of place.

30. Some suggestions for Fig 9:

a. 9b bears no relation to 9a and should be a standalone figure.

b. The rightmost column in 9a should be labeled as its own subplot.

c. Instead of the 60 degrees inset on the side of 9a, can you show orientation in terms of the yellow-green-blue color bar? I’m imagining a color fan or a series of colored arrows that allow the reader to easily visually compare, say, 57 vs. 67 degrees.

d. Another subplot between the current 9a and 9b could show the (clustered AR days) – (all AR days) anomaly, which is arguably more interesting than the NDJ-JFM anomaly and would give visual support to the statement at Lines 287-289.

31. Figs 9a & 10: can you tell me in the figure caption (and in the text) which orientation anomaly is positive and which is negative? Maybe a clockwise/counterclockwise arrow, another inset to the right of 9a, or text stating that a positive anomaly is more northerly vs. negative is more westerly, etc. Right now I have no idea which is which.

32. Line 366-368: You could say “we consider six dominant climate modes at the daily scale” to avoid the redundancy of specifying that every index is daily. Also, what makes these climate modes dominant?

33. Line 374: Link does not open, 404 error.

34. Line 374-375: Why is ENSO signal removed from PNA, and why is it not removed from any other indices?

35. Line 386-389: It’s not clear what “these climate modes of variability” and “these climate indexes” are referring to. Also the MERRA-2 AR events dataset has not been discussed yet. It seems odd to have this sentence occur before all climate mode datasets have been introduced and to have a paragraph break separating PDO from all other indices.

36. Line 387 & 397: Consider defining the time overlaps in terms of water year (starting in October) rather than the 1940/1941 notation used here for clarity.

37. Line 398-401: Okay, now I understand why PDO is in a separate paragraph. It would be much better to have this information at the start of this section, maybe at Line 369. It would also be helpful to have a pointer to the next section, where you explain why you are using two reanalysis products (i.e., capturing short- vs. long-term climate variability)

38. Line 405-407: Sentence fragment

39. Line 416: MERRA-2 is at a 3-hour timestep, not 6 hours. ERA5 is at an hourly timestep, not 6 hours.

40. Line 423-434: “long-period climate modes variability PDO” feels awkward. Is there a missing/extra word in here?

41. The Guan & Waliser algorithm does not inherently have object IDs, right? How long does the gap of non-AR timesteps have to be to identify a unique AR event? Can multiple unique AR events occur on the same day? Why do you even need to calculate AR days, when all of the results in Figs 3-8 are measured in terms of unique AR events?

42. Line 446: The variability does not alternate between positive and negative phases, the modes themselves do. “Climate modes of variability” is not the same as “variability of climate modes.”

43. Lines 450-454: I have several questions about this methodological choice. When you say “all possible combinations,” what do you mean? Did you include interaction effects? To what degree? Did some coefficients get left out of the model? What regression forms were tested and which were rejected? What is the form of the final regression model?

44. Line 454: The package name is “survival”, not “Survival”; case matters in R.

45. Lines 459-464: It is repetitive to have the same information twice, once in parentheses and once in a second sentence. I would keep one or the other, and given the choice I would remove the parentheses and keep the second sentence.

46. Line 466: Why would you average orientation over AR days when it could be averaged over AR events? Again, why are AR days calculated at all? It is just as simple and more accurate to count the event starts within a 7-day interval.

47. Line 470: If the one-week intervals are fixed, then the results will be different based on the start day; for example, the third AR in Fig 9b would be in a cluster if the one-week window started two days later. Why was the choice made to define temporal clusters in this way, when multiple other definitions of temporally compound ARs exist, and have you tested the effect of the fixed interval start date on your results?

48. Line 473: If I understand correctly, you are comparing the mean orientation of all ARs to the mean orientation of the subset of temporally clustered ARs. I’m not sure how or why calculating the “mean IVT orientation throughout their life cycle” is related to this. Are you averaging twice, once per cluster and once per grid cell? Could you streamline your methods (and

your description) to only average once?

49. Fig 10: What anomalies are we looking at? Is this temporally clustered AR orientation vs. all AR orientation? If so, please state that in the text and in the figure caption.

50. Lines 295-315: Again, there are a bunch of results without much interpretation. Also why did you highlight the modes you highlighted? All other plots have gotten discussion of every mode (which may be overkill, but it is the precedent you've set).

51. Please list the climate modes in the same order throughout the paper (including the figures and figure captions).

52. Line 324: Not necessarily true; see comment about results coming from different regressions.

53. Line 331-339: Tell me why each of these findings matters! "This pattern is interesting/unexpected novel because... This is consistent with/differs from other work in that..."

54. Line 334: Please relate this back to which directions have been found to have the largest hydrologic impacts in the western US.

55. Line 342-343: Why is MJO highlighted again here?

56. Line 347-349: Thank you for connecting your results to previous work. I would love to see more of this throughout the results and conclusion.

57. Lines 340-362: I suggest (a) combining the paragraphs at Lines 340 and 344 and (b) separating the discussion of AR orientation starting at Line 350 from the discussion of forecasting starting at Line 355. The last section of text makes an important point about the value of this work (although it never ties in seasonality), but needs a bit more filling out to be a really strong ending.

58. Line 509: It would be far better if the analysis code (or at least a demonstration script reproducing the main figures) was publicly available via Github or another platform. The analysis and results of this paper are not reproducible as is.

59. General figure comments:

a. Labels and axes are barely legible and should be in a much larger font. Figs 2, 5, 6, 8, 9a, and 10 are particularly bad.

b. Please make the significance dots larger as well.

Reviewer #2 (Remarks to the Author):

Summary:

Clustering of ARs has profound economical implication as is evidenced in the recent literature. AR orientation is another important factor that is tied to the extent of economic damages. The main idea of this manuscript is the early winter versus late winter influence of six climate modes on temporal clustering and orientation of ARs. The result suggest the AR clustering is modulated by climate modes, AO, PNA etc. and that the extent of modulation varies with the progression through the winter season, for example, QBO influence on early winter clustering is nearly negligible, but in late winter period, there is significant of QBO influence on clustering over west coast.

Based on my holistic assessment of the manuscript, several important inferences can be made from the results, which can be critical for future understand and prediction of ARs over the west Coast. However, there are certain aspects of the manuscript where I think the authors can make improvements-- these are given in my comments below. The manuscript has significant potential for advancing science of ARs, thus I would strongly encourage the authors to revise it for acceptance in the Journal.

Major Comments:

1. While the authors provide compelling literature on evidence of AR clustering (most of the references from 6-17), we may add more discussion on the causes of clustering hypothesized in these references. It is necessary to give a clear and state - of -the -art background on the research question, which is related to the causes of AR clustering; apart from lines 66-69, I did not find literature on the AR clustering causes. I would be better to first establish background on the current understanding of the causes of AR clustering and then build on that to present the research question targeted in the manuscript.

2. Figure2: Some similarity in the influence of the climate modes can be observed here. For example, notice the -ve effect of AO and +ve effect of PDO over much of the region near the coast. Then one would wonder: are AO and PDO somehow correlated here?

3. L160-177: In Figure 3 and throughout the manuscript, We discuss beta that represents the extend or magnitude of clustering; while as, in Figure 3 and others, we provide number/frequency of ARs as an evidence of clustering. However, clustering extent and frequency of ARs, though related, are two separate concepts. It is possible to have high number of ARs but no clustering, since the ARs may have occurred at regular intervals. I think the authors may try to adopt some better alternative for showing evidence of clustering (Figure 3 and other). One way would be to consider time between events or its variance as a measure of clustering extent.

4. Figure 9a: Comparing top and bottom panels, we can see that unique clustered ARs are more southerly than all AR days; we may note this and try to discuss what may be the cause. Southerly orientation of unique clustered ARs and westerly orientation of all ARs days indicate that over their lifetime, ARs shift from southerly to westerly. This implication is based on the definition of unique clustered ARs shown in Figure 9b. We may think about this.

5. L439: Isn't this Poisson process for each day? If yes, then what is point of naming it as Cox process? Writing the complete

probability density function here would be useful for better interpretation. why does t range from 1 to 365 when we only consider winter AR?

Minor Comments:

Figure3: In regions where frequency is nearly zero, how did the authors tackle the most-often zero issue?

Figures 4&5: What is the underlying null hypothesis at each gridcell for which statistical significance of estimated?

Figures 3 & 7 seem redundant; we may keep Figure 7 only.

L255-260: Again Figures 7 & 8 seem redundant.

L45: All of the 1-5 references are not on flood damages, please verify.

L46: Again, all of the 6-9 references do not appear related to the topic of discussion; reference 10 is not on ARs.

L47: please elaborate what is meant by indicator here.

L58-59: if they occur in bursts, the by definition they are dependent, right?

L70: Does this mean rate of AR clustering is more than clustering at random?

L82: We may keep either "mean" or "primarily" not both.

L91: This doesn't appear to be linked with relationship between AR orientation and extreme precipitation, which is the main theme of the paragraph.

L287-289: I wonder why this may be the case?

L374-375: This is important; please explain in detail.

L403-404: At other instances, we say AR catalog from 58, here we are detecting ARs. If we already have an AR catalog, then lines 403-414 may be omitted.

L426-427: how is independence defined?

L435: So the events are not independent, right?

Best wishes

Munir Ahmad Nayak

Communications Earth & Environment is committed to improving transparency in authorship. As part of our efforts in this direction, we are now requesting that all authors identified as 'corresponding author' create and link their Open Researcher and Contributor Identifier (ORCID) with their account on the Manuscript Tracking System prior to acceptance. ORCID helps the scientific community achieve unambiguous attribution of all scholarly contributions. You can create and link your ORCID from the home page of the Manuscript Tracking System by clicking on 'Modify my Springer Nature account' and following the instructions in the link below. Please also inform all co-authors that they can add their ORCIDs to their accounts and that they must do so prior to acceptance.

Version 1:

Decision Letter:

Dear Dr Yang,

Your manuscript titled "Temporal clustering of unique atmospheric rivers impacting the Western U.S.: seasonality and climate modes modulation" has now been seen by our reviewers, whose comments appear below. In light of their advice we are delighted to say that we are happy, in principle, to publish a suitably revised version in Communications Earth & Environment.

We therefore invite you to revise your paper one last time to address the remaining concerns of our reviewer 1. At the same time we ask that you edit your manuscript to comply with our format requirements and to maximise the accessibility and therefore the impact of your work.

EDITORIAL REQUESTS:

****Please take care to match our formatting and policy requirements. We will check revised manuscript and return manuscripts that do not comply. Such requests will lead to delays. ****

SUBMISSION INFORMATION:

OPEN ACCESS:

Communications Earth & Environment is a fully open access journal. Articles are made freely accessible on publication. For further information about article processing charges, open access funding, and advice and support from Nature Research, please visit <https://www.nature.com/commsenv/open-access>

Link Redacted

Best regards,

Alireza Bahadori, PhD
Associate Editor
Communications Earth & Environment

REVIEWERS' COMMENTS:

Reviewer #1 (Remarks to the Author):

Thank you to the authors for their edits to the manuscript and their thoughtful responses to the comments. I really appreciate the additional interpretive text that has been added throughout, and I like the symmetry now between the way temporal clustering + orientation are discussed in the introduction. I have two remaining items of concern, as well as a few editorial comments below. I believe that this article is ready for publication with minor revisions.

First, Fish et al. (2019, 2022) use a five-day instead of a seven-day window, and they use an algorithm that starts counting a cluster/family based on some threshold criteria rather than using fixed intervals. To be clear, I think all of these choices made in this manuscript are defensible (and thank you for testing the sensitivity of the start of the fixed window), but it is incorrect to state that the choices are directly supported by the cited literature.

Second, I do not understand the rationale for comparing clustered AR events against all AR days in Fig 9. As mentioned in both the authors' comment responses (30d) and in the manuscript itself (Line 468), this is not an apples-to-apples comparison, because the orientation of long vs. short clustered ARs is weighted equally but a long AR would be weighted more heavily than a short one in the mean orientation of AR days. The effects of clustering and duration are therefore lumped together with no way to distinguish the contributions of each individually. It would be more consistent to compare either (a) all days within clustered ARs vs. all AR days or (b) clustered unique AR events vs. all unique AR events. Considering that analysis (b) is already done in Fig S2, I believe that it belongs in the main text in place of Fig 9.

Additional comments:

- Line 134: run-on sentence
- Line 141: great addition
- Line 193: move "and" to last item in list
- Line 225: should this be "temporal clustering of unique ARs"?
- Line 229: it would be helpful to move some the contextual description about what the Cox regression measures, what the coefficient beta measures, etc. from the Methods up to here
- Line 275: should be present tense
- Fig 9:
 - o The labels of 9a and 9b seem to be flipped, please check in-text references to both
 - o It would be helpful if Fig 9 were organized in the same order it was described in the text: first the schematic, then the top row of (b), then the bottom row of (b), etc.
 - o Why does Fig 9(c) have a different color scale than 9(d) for the same values?
 - o Please mention the orientation diagram 9(e) somewhere in text
- Lines 467 & 471: number supplementary figures
- Line 517 "in opposing ways" – relative to what?
- Line 549: ARs? Or clustered ARs?
- Line 796: typo
- Line 826: typo

Reviewer #2 (Remarks to the Author):

Dear Editor

Sorry for the delayed review. I'm happy with the revision made by the authors.

Thank you

Dear Editor and Reviewers,

We gratefully appreciate your insightful comments on our manuscript, which we hope have improved the quality of this work. Please see our responses below (in blue) to each comment (in black).

Sincerely,
Zhiqi Yang

Response to Reviewer 1's comments on:

Temporal clustering of unique atmospheric rivers impacting the Western U.S.: seasonality and climate modes modulation

ZHIQI YANG*, MICHAEL J. DEFLORIO, AGNIV SENGUPTA, JIABAO WANG, CHRISTOPHER M. CASTELLANO, ALEXANDER GERSHUNOV, KRISTEN GUIRGUIS, EMILY SLINSKEY, BIN GUAN, LUCA DELLE MONACHE, F. MARTIN RALPH

Reviewer #1 (Remarks to the Author):

This paper looks at the effects of climate modes and early vs. late winter seasonality on both temporal clustering and IVT orientation in ARs affecting the North Pacific Ocean and Western US. While many of these factors have been examined separately in the AR literature, this is the first to look at interactions between them. The authors have a clear understanding of the existing literature, the analysis is strong, and overall this paper would be a positive contribution to the field.

My main issue with this paper is the complete lack of interpretation. In many cases the authors simply list differences or patterns in the figures without contextualization or comparison, and while the “what” and “how” were well described, I was often left missing the “why,” which left the paper feeling much more like a disjointed collection of facts than a cohesive narrative. I also have some concerns with the metrics of temporal compounding, as detailed in my comments below. I recommend minor revisions.

Response:

We want to thank the Reviewer for her/his comments and suggestions, which we have addressed point-by-point below.

Additionally, we have incorporated more literature to explain the potential physical reasons behind the influence of climate modes, including why the PNA shows a strong signal and the effects of the AO, MJO. We have also referenced work on AR families (Fish et al. 2019, 2022) to justify our use of a 1-week window in the composite analysis and to provide evidence supporting

the reasonableness of these metrics. It is also related to our desire to examine subseasonal predictability of temporal AR clusters in future work. Weekly aggregates (or two-week aggregates) of variables in subseasonal predictability studies are commonly used. We added more details in following responses.

Fish, M.A., et al., Large-scale environments of successive atmospheric river event leading to compound precipitation extremes in California. Journal of Climate, 2022. 35(5): p. 1515-1536.
Fish, M.A., A.M. Wilson, and F.M. Ralph, Atmospheric river families: Definition and associated synoptic conditions. Journal of Hydrometeorology, 2019. 20(10): p. 2091-2108.

Major comments:

1. Throughout the paper, the work on AR orientation is really framed as an afterthought, not a main contribution. The abstract includes only one of the two central questions you pose in the introduction (and I don't like the repetition of "yet the enigma persists" in both the abstract and the body of the paper). It is not until the very last paragraph of the conclusion that you mention why you are examining both: i.e., both are major determinants of AR impacts and climate modes can modulate the two in different (and sometimes opposing) ways. AR orientation deserves a more complete treatment in the abstract, introduction, and interpretation of results.

Response:

Thank you for your suggestions. We have expanded the explanation of AR orientation in the abstract, introduction, and results sections to ensure it receives more comprehensive treatment throughout the paper.

For example, in the abstract we now state (see lines 32-33) "AR orientation also plays a crucial role in modulating hydrological impacts through terrain interaction". In the abstract we also mention our focus relating climate modes including ENSO to AR orientation (see lines 38-39). We have also eliminated the phrase "yet the enigma persists" in the abstract. Throughout the revised manuscript, we have added discussions specifically regarding orientation (for example, see lines 47-49, 55-56, 82-102, 265-290).

2. I would encourage the authors to try to, in their own words, answer "This finding is important because..." for the discussion of every single figure. The stated goal of this paper is to improve seasonal-to-subseasonal forecasting; how do different composite patterns support that goal? What can we learn about anticipated impacts? What does it mean for forecast skill when modes have similar vs. contrasting anomaly patterns? There is a lot of information in these figures; your goal is not to summarize it all, but to extract the key insights and tell the reader what to care about.

Response:

Thank you for your thoughtful feedback! We appreciate the suggestion and framed our discussion around the importance of each figure in relation to our overall goal of improving seasonal-to-subseasonal forecasting.

In response, we revised the results sections to explicitly address why each finding is important. By focusing on these key insights, we aim to provide a clearer narrative that guides the reader through the most critical aspects of our findings and their relevance to improving AR forecasting. We added more text for Figures 2, 3, 4, 6, 7, 9, and 10.

For example, for Figures 2, 3, 4,

‘Therefore, these findings highlight how different climate modes modulate AR temporal clustering. By incorporating the influence of large-scale climate modes and considering temporal clustering, they enable more accurate S2S forecasting of AR events (similar example: hurricane forecast improvement [44]).’ See Lines: 157-160

‘These results offer an independent approach from the previous paragraph, highlighting how climate modes modulate temporal AR clustering and showing the variation in the number of unique ARs associated with the positive and negative phases of climate modes within a one-week window. Consequently, we can improve prediction of the number and impact of ARs based on the current phase of these climate modes.’ See Lines: 176-180

‘Since MJO phases are an important source of S2S predictability, this finding offers valuable insights into the expected number of ARs based on the current MJO phase, thereby improving forecast accuracy.’ See Lines: 193-195

[44] Jagger, T.H. and J.B. Elsner, Hurricane clusters in the vicinity of Florida. Journal of Applied Meteorology and Climatology, 2012. 51(5): p. 869-877.

3. The authors assume that the number of unique ARs in a 1-week window can be used as a measure of temporal compounding. I am not convinced that this is an appropriate or accurate metric. Many of the paper’s main results (Figs 3-8) depend on this logical leap, so further explanation and support of this choice is required.

Response:

We appreciate your feedback on the use of a 1-week window to measure temporal compounding of ARs. We acknowledge that this metric may appear to be arbitrary, and we agree that further clarification is warranted. The primary rationale for using a 1-week window is to establish a foundational framework for future research on subseasonal AR clustering and prediction within a dynamic statistical forecasting framework. While a longer window could offer additional insights, our approach provides a starting point for understanding temporal compounding on a manageable scale.

To support our choice of a 1-week window, we have referenced relevant studies on AR families (Fish et al. 2019, 2022), which classify AR objects as an AR family if two or more AR objects occur within the 120-h aggregation period. These references provide evidence that this metric is reasonable and appropriate for capturing clustering patterns in AR activity.

Additionally, in the Cox regression analysis, we do not explicitly set a window to identify temporal clustering. Instead, the significant β coefficient indicates the presence of temporal clustering among unique ARs, independent of the 1-week window setting. Therefore, this specific window does not influence the results of temporal clustering.

We incorporated additional discussion in the revised manuscript to address these concerns and strengthen the rationale for our chosen metric. Thank you for bringing this important point to our attention. In the revised manuscript, we have added the following text in Methods to clarify (see lines 525-532).

‘Setting number of unique ARs anomalies in a 1-week window

We use of a 1-week window to measure temporal compounding of ARs in composite analysis, which is to establish a foundational framework for future research on subseasonal AR clustering and prediction within a dynamic statistical forecasting framework. The 1-week window also based on research of [13, 14], which classify AR events as an AR family if two or more AR objects occur within the 120-h aggregation period. Therefore, 1-week is an appropriate window to identify clustering. While a longer window could offer additional insights, our approach provides a starting point for understanding temporal compounding on a manageable scale.’

Fish, M.A., et al., Large-scale environments of successive atmospheric river event leading to compound precipitation extremes in California. Journal of Climate, 2022. 35(5): p. 1515-1536.
Fish, M.A., A.M. Wilson, and F.M. Ralph, Atmospheric river families: Definition and associated synoptic conditions. Journal of Hydrometeorology, 2019. 20(10): p. 2091-2108.

4. Fitting two Cox regressions, where one regression considers five modes and the other considers only PDO, means that the coefficients from the first regression are not directly comparable to those from the second. It is misleading to put them on the same plot with the same color scale without a note acknowledging this expected difference. It also means any comparative interpretation of the magnitude of the coefficients is flawed. Very little information is provided about the regressions, and more detail is needed about the regression forms that were tested for AO, PNA, MJO, QBO, and ENSO.

Response:

Thank you for your suggestions. We agree that the coefficients from the first regression are not directly comparable to those from the second. We have added a note in both the text and the figure captions to clarify this.

See lines:138-140. *‘Note that PDO and the other five modes are analyzed in separate regressions, making direct comparison of PDO coefficients with the second regression not possible (see Methods).’*

See lines: 509-512 *'Noted that since we use two separate Cox regression analysis for AO, QBO, ENSO, MJO, PNA and PDO, the coefficients from the first regression (i.e. AO, QBO, ENSO, MJO, PNA) cannot directly be comparable to those from the second (i.e. PDO).'*'

See Figure 2: figure captions *'Noted that the coefficients from the regression of AO, QBO, ENSO, MJO, PNA cannot directly be comparable to those from PDO (see Methods).'*'

Additionally, we have removed any comparative interpretation of the magnitude of the coefficients between the PDO and the other five climate modes.

Furthermore, we have included more details about the regression models for AO, PNA, MJO, QBO, and ENSO in the Methods section to provide better context. Specifically, to consider the most significant covariates in each grid cell, we cover all possible combinations (i.e., 32 combinations in total, calculated by permutations and combinations $C(5,0)+C(5,1)+C(5,2)+C(5,3)+C(5,4)+C(5,5)$) of AO, QBO, ENSO, MJO, and PNA (Yang, Z., & Villarini, G. (2021)). For instance, the combined model includes combinations like:

- ~AO
- ~AO+QBO
- ~AO+ENSO
- ~QBO+ENSO+MJO
- ~AO+QBO+ENSO+MJO

...

Note that the final regression model may be different in different grid cells.

See lines: 500-508. *'In this study, to consider the most significant covariates in each grid, we cover all possible combinations (i.e., 32 combinations in total, calculated by permutations and combinations) of AO, QBO, ENSO, MJO, and PNA. We also took into account the correlation among the different predictors in the context of collinearity. Anderson and Burnham (2004) [68] recommended using a cutoff value of |0.95| for excluding predictors, while retaining those with seemingly significant correlations. In this study, the correlation values among these AO,PNA,MJO,QBO,ENSO are generally much smaller than |0.3|. Therefore, we do not believe that collinearity among the climate indices has impacted our results. Then, to find the best Cox regression model, we select a regression model with the smallest Akaike information criterion (AIC [69]).'*

* The correlation coefficients of all pairs of AO, PNA (after remove ENSO),MJO,QBO,ENSO:
Please see Table S1

Climate Modes	NDJ	JFM	NDJFM
AO,PNA	-0.2001486018273725	-0.24637864819105046	-0.22586641135570698
AO,RMM1	0.1209974433901579	0.13668287155888773	0.12040836802640792
AO,RMM2	-0.14273152700488279	-0.09281873451330258	-0.11016684247253844
AO,QBO	0.09406993774545983	-0.002884660187138659	0.04219392930520683
AO,ENSO	0.04970780835261645	-0.05224780847040277	0.017161228297032597
PNA,RMM1	-0.12115710713056552	-0.21255015548629774	-0.14734450058746087
PNA,RMM2	0.011772280426781672	0.06678823179269228	0.06126415754558458

PNA,QBO	-0.004741184607769526	0.04540358271568377	0.04404277598393147
PNA,ENSO	-6.121725906730966x10 ⁻⁵	0.0012393480442218633	0.07669100607194652
RMM1,RMM2	-0.026638671121564098	-0.10424254972073081	-0.08375252047633747
RMM1,QBO	0.06267201328652588	0.0036516165803568298	0.018184691699926075
RMM1,ENSO	0.028306267340458997	-0.011848280923704407	0.0160019803792233
RMM2,QBO	-0.05533665225368805	-0.08861361561655196	-0.06668185623789367
RMM2,ENSO	-0.13531087401849876	-0.01258077309207827	-0.10997883961040632
QBO,ENSO	0.18525855802209815	0.14935683005025127	0.15679703260914

Thank you for bringing this to our attention!

Minor comments:

5. Line 57: should be “AR families”, not family

Response:

We rephrased this sentence in the revised manuscript. Thank you.

6. Line 58-59: This seems circular; the presence of clustering suggests clustering? Also, it is awkward to define “independent ARs” at the same time as defining ARs within bursts.

Response:

Thank you. here we want to say unique ARs; each of them has different track IDs. We rephrased this sentence in the revised manuscript.

‘Here, we consider unique AR events (or, in short, ‘unique ARs’); for a unique AR event, if it occurred on consecutive days, only the first day is counted. Unique AR events exhibiting temporal clustering generally show alternation between quiet and active periods regarding the number of events.’

7. Line 71: “temporally clustered ARs”, not AR

Response:

We rephrased this sentence in the revised manuscript.

8. Line 74: Likely do not need all these citations; 19-20-21 seem very similar, as do 22-23-24-25-29. Also “floods on the Potomac River” is a bit jarring because it is so much more geographically specific than the others mentioned and it is a measure of impact rather than hazard. Something like “streamflow in the eastern US” would be more in line with the way you discuss the other references.

Response:

We removed some of these citations (original 20-22-23-29) to make it concise and rephrased this sentence focusing on streamflow in the eastern US in the revised manuscript.

‘Previous research has also explored temporal clustering in other hydrometeorological variables, such as streamflow in the eastern U.S., heavy precipitation, and hurricanes in the

Central U.S., Europe, and China [5, 14-19] as well as the ability of reanalysis data and climate models (GCM/RCM) to capture these clustering characteristics [19-21]. These studies highlight the significant influence of large-scale atmospheric circulation on the temporal clustering of hydrometeorological variables.'

9. Line 80: Add a citation number for Slinsky et al.

Response:

We added the citation number in the revised manuscript.

10. Line 78-104: This could be set up better.

a. Please streamline the citations more. The paragraph feels very repetitive (“Paper A said this. Paper B said this. Paper C...”) without a clear thread. What should my takeaways be—the most common/most impactful orientation by global region? Or just the fact that impacts are influenced by orientation? If the goal is the former, please state the takeaways explicitly; if it is the latter, you could make the point without nearly as much distracting detail.

Response:

Thank you for your suggestions! We appreciate your feedback on improving the structure and clarity of this section. Our goal is to highlight the common AR IVT orientation in the Western U.S. and how hydrological impacts are influenced by orientation. For example, extreme streamflow and intense precipitation in coastal Northern California are primarily associated with more southerly AR IVT orientation, while extreme streamflow and intense precipitation in Washington and Oregon are primarily associated with more westerly AR IVT orientation, which is influenced by whether AR hits the terrain more vertically.

We rephrased this paragraph to make it clear in the revised manuscript.

See lines: 82-102. *'Moreover, the orientation of ARs has distinct effects on ... this will allow for more accurate estimation of subsequent hydrological impacts and significantly benefit water management efforts in the Western U.S.'* entire paragraph.

b. The entire introduction has focused on temporal compounding up to this point, with no previous mention of orientation, so the transition between this paragraph and the preceding one is a little abrupt.

Response:

Thank you for your suggestion. To ensure a smoother transition, we have revised the introduction to include orientation-related content in the first paragraph. This addition should help the discussion flow more naturally from the topic of temporal compounding to orientation.

c. The connection between [32] and orientation is unclear.

Response:

We rephrased this sentence and combined it with other references in the revised manuscript to highlight the relationship between AR orientation and extreme precipitation.

'South-southwesterly ARs are associated with more precipitation and stronger moisture transport in Northern California, while westerly orientations are linked to extreme streamflow and intense precipitation in Washington and Oregon, influenced by complex topography [24] [6] [25].'

[6] Hu, H., Dominguez, F., Wang, Z., Lavers, D. A., Zhang, G., & Ralph, F. M. (2017). Linking atmospheric river hydrological impacts on the US West Coast to Rossby wave breaking. *Journal of Climate*, 30(9), 3381-3399.

d. The description of [33] is a run-on sentence.

Response:

Thank you. We rewrote this sentence in combination with conclusions from other literature in the revised manuscript.

See lines: 88-91. *'South-southwesterly ARs are associated with more precipitation and stronger moisture transport in Northern California, while westerly orientations are linked to extreme streamflow and intense precipitation in Washington and Oregon, influenced by complex topography [24] [6] [25].'*

[25](original [33]) Neiman, P.J., et al., *Flooding in western Washington: The connection to atmospheric rivers. Journal of Hydrometeorology*, 2011. 12(6): p. 1337-1358.

e. For [9], where is the Teifi catchment?

Response:

Teifi and Dyfi are two impermeable catchments on the west coast of Britain. Dyfi is more north, and Teifi is more south. We rephrased this sentence in the revised manuscript to make it clear.

'In other regions, west-southwesterly orientations yield the most pronounced hydrological responses in Britain's Dyfi catchment [26].'

f. Line 101: This is really important! This is a critical connection that explains why an examination of orientation has value for forecasting, and it should be highlighted accordingly.

Response:

Thank you for your suggestions! We revised the text to ensure that this point is highlighted more prominently to emphasize its importance.

See lines: 93-97. *‘At subseasonal lead times, AR orientation is crucial, as some orientations are much more predictable in the Western U.S. and have significant impact implications [27]. This critical insight underscores the need for targeted forecasting strategies that account for these variations in predictability, making it an essential consideration in hydrological predictions and risk assessments.’*

[27] DeFlorio, M.J., et al., *Experimental subseasonal - to - seasonal (S2S) forecasting of atmospheric rivers over the western United States. Journal of Geophysical Research: Atmospheres, 2019. 124(21): p. 11242-11265.*

11. Line 105: I do not think you actually answer your two main motivating questions as posed, so it would be helpful to both you and the reader to (a) make them more specific and (b) explicitly state the value of answering each.

Response:

Thank you for your suggestions. We rephrased the questions and added values in the revised manuscript.

See lines:77-81. *‘These studies highlight the significant influence of large-scale atmospheric circulation on the temporal clustering of hydrometeorological variables. Yet, the enigma persists: what is the interacting effect of climate modes and seasonality on AR temporal clustering? Answering this question is crucial as it will yield new approaches and predictors for improving the forecasting of temporal AR clustering and related economic losses at subseasonal-to-seasonal timescales.’*

See lines:97-102. *‘Therefore, certain orientations of ARs can exacerbate flood severity, particularly when they occur in succession over a short period of time. So, what is the interacting effect of climate modes and seasonality on AR orientation? Addressing this question is essential, as it will enhance the forecasting of AR orientation. When combined with improved forecasting of temporal AR clustering, this will allow for more accurate estimation of subsequent hydrological impacts and significantly benefit water management efforts in the Western U.S.’*

a. “Why do ARs cluster in time?” ◊ what is the interacting effect of climate modes and seasonality on AR clustering? (it is important to answer this question because...)

Response:

We have rephrased the question to “What is the interacting effect of climate modes and seasonality on AR temporal clustering?” and added that it is important to answer this question because it will provide a new approach and predictors for improving the forecasting of temporal AR clustering and related economic losses at subseasonal-to-seasonal timescales.

b. “What are the characteristics of the AR orientation associated with the temporally clustered ARs?” ◊ what is the interacting effect of climate modes and seasonality on AR orientation? (it is important to answer this question because...)

Response:

We have rephrased the question to “what is the interacting effect of climate modes and seasonality on AR orientation?” and added that it is important to answer this question because it will enhance the forecasting of AR orientation. When combined with improved forecasting of temporal AR clustering, this will allow for more accurate estimation of subsequent hydrological impacts and significantly benefit water management efforts in the Western U.S.

12. Line 114: two “and”s

Response:

We rephrased this sentence in the revised manuscript.

13. Line 115-116: Incomplete clause. Should it be seasonal disparities “exist” between?

Response:

We rephrased this sentence in the revised manuscript.

14. Line 118-125: It’s not really the analysis of AR clustering that accomplishes this, it’s the discovery of relationships between factors with high subseasonal predictability and AR clustering/orientation.

Response:

Thanks for your suggestions. We agree that this paper emphasizes the relationship between subseasonal predictors and AR clustering/orientation, which provides a basis for future subseasonal AR clustering predictability. We rephrased this sentence in the revised manuscript.

‘Furthermore, the discovery of relationships between factors with high subseasonal predictability and AR clustering/orientation would lay a foundation to evaluate the subseasonal (2-6 week lead) and seasonal (3-6 month lead) predictability of AR sequences occurring within a short time period.’

15. Line 134: The phrasing “by around” is confusing. I think you mean “to around”; otherwise it reads like the number of unique ARs along the coastline is $1.4 - 1 = 0.4$. Also they should all be in units of “ARs per week”, not “AR per week”

Response:

We rephrased this sentence in the revised manuscript.

16. Fig 2: Column and row labels would be helpful.

Response:

We revised the Figure 2 as suggestions.

17. Line 137-159: This is such a disjointed paragraph. Can you add interpretation—how does this compare to previous literature? Why do these specific spatial patterns matter?

Response:

Thanks for your suggestions. We added more literature comparisons.

See lines:140-142. *‘Specifically, the AO influences Northern California, with more unique AR clustering during its negative phase, consistent with Guan et al. [41].’*

See lines:147-151. *‘When PDO is in a positive phase, unique ARs clustering is higher over California, while from Washington/Oregon to the Rocky Mountains, unique ARs clustering is lower, and vice versa for negative PDO, aligning with Gershunov et al. [43], demonstrating that the PDO significantly modulates AR frequency over West Coast.’*

See lines: 152-156. *‘The PNA signal shows a northwest-southeast pattern over the Western U.S. The PNA’s positive phase increases unique AR clustering in the Northwestern U.S., while its negative phase does so in the interior Southwestern U.S., as noted by Zhou et al. [13] in investigating PNA-like pattern related to AR clustering.’*

These specific spatial patterns matter since *‘Each of these climate modes significantly modulates precipitation and AR activity in the Western U.S. [23, 51]. We consider these climate modes dominant based on the literature review of their significant impacts on atmospheric rivers and precipitation over the western U.S. and on their utilization in the S2S operational forecasts. For instance, Guan and Waliser [41] found that the AO, ENSO, MJO, and PNA significantly impact AR frequency and precipitation in the western U.S. Specifically, the AR activity over California was linked to the negative phase of both the PNA and AO. ENSO modulates AR frequency along the western U.S. coast. The AR activity in the California Sierra Nevada is shown to be significantly strengthened when MJO convection is in phase 6 (Guan et al. [52]). The PNA-like pattern strongly correlates with temporal AR clustering in the western U.S. (Zhou et al. [13]). In the operational forecast, MJO and QBO strongly modulate subseasonal AR activity and precipitation in California, with reduced AR activity during easterly QBO and increased activity during westerly QBO in the cold season (Castellano et al. [36]). Additionally, Gershunov et al.*

[43] reported that AR activity along the west coast of North America generally increases during positive phases of the PDO and decreases during negative phases.’ (See lines: 402-416).

We added the importance of these spatial patterns in the Method section and pointer at the beginning of the paragraph. See lines: 133-135. *‘Then, we investigate the unique ARs temporal clustering within the extended boreal winter over the Western U.S., and the impact of AO, QBO, ENSO, MJO, PDO, and PNA (see Methods) on the magnitude and spatial structure of the clustering.’*

a. Why is PNA signal so much stronger than the others?

Response:

PNA is a strong mode of climate variability affecting Western North America, particularly affecting the Western U.S. Its strength relative to other climate signals is due to its direct influence on these regions, making its impact more pronounced. We added more explanation in the revised manuscript.

See lines: 156-157. *‘PNA shows a strong impact as it is a mode of climate variability affecting Western North America, particularly affecting the Western U.S.’*

b. Line 157: This is the topic sentence for this paragraph, it would help a lot to move to the top of the paragraph so the reader knows what to expect.

Response:

Thank you. We rephrased this paragraph as suggestion.

18. As a general comment, the (over)use of “specifically”, “furthermore”, “moreover”, etc. between the results of different climate modes is distracting. Transition words are not necessary between elements of a list.

Response:

We have rephrased these sentences in the revised manuscript and removed some transition words in the results of climate modes to make it clearer and more compact.

19. Line 160: “the number of unique ARs”: are you counting AR days or AR events? If events, are you counting starts?

Response:

We are counting AR events. Yes, we count only the start of unique AR events. If ARs occur on consecutive days, we consider only the first day of the event. We defined unique AR events clearly in the introduction section of the revised manuscript. Thank you.

20. Line 175-176: This is a helpful global observation that belongs at the top of the paragraph, before all of the details associated with specific modes. Similar to above, please provide

interpretation for each climate mode (“This pattern is interesting/informative/noteworthy because...”)

Response:

Thanks for your suggestions. We moved this summary sentence to the top of the paragraph and added more interpretation for each climate mode in the Methods section.

21. Line 178: Why, out of all the climate modes, does MJO deserve a closer examination?

Response:

The MJO deserves closer examination because previous studies have demonstrated that the MJO not only modulates AR activity, but also contributes to increased predictability of large-scale dynamics and moisture transport on subseasonal timescales.

For instance, DeFlorio et al. (2019) and Zhang et al. (2023) found MJO phases significant modulations of number of 1-week AR occurrences and AR integrated water vapor transport (IVT) forecast skill at S2S timescales. Castellano et al. (2023) explored the combined use of MJO phase and QBO phase as a predictor to improve subseasonal forecasts of AR activity in California.

Also, Toride and Hakim (2022) used RMM1 and RMM2 to identify the connection between MJO events and AR activity over the Pacific Northwest coast. Tseng et al. (2021) used RMM1 and RMM2, indicating that the dynamics of the MJO determine most of the predictable signals on subseasonal timescales, such as ARs.

Therefore, we included both MJO indexes (RMM1/RMM2) and MJO phases in the study.

*DeFlorio, M. J., Waliser, D. E., Guan, B., Ralph, F. M., & Vitart, F. (2019). Global evaluation of atmospheric river subseasonal prediction skill. *Climate Dynamics*, 52, 3039-3060.*

*Castellano, C. M., DeFlorio, M. J., Gibson, P. B., Delle Monache, L., Kalansky, J. F., Wang, J., ... & Anderson, M. L. (2023). Development of a statistical subseasonal forecast tool to predict California atmospheric rivers and precipitation based on MJO and QBO activity. *Journal of Geophysical Research: Atmospheres*, 128(6), e2022JD037360.*

*Zhang, Z., DeFlorio, M. J., Delle Monache, L., Subramanian, A. C., Ralph, F. M., Waliser, D. E., ... & Lin, H. (2023). Multi-Model Subseasonal Prediction Skill Assessment of Water Vapor Transport Associated With Atmospheric Rivers Over the Western US. *Journal of Geophysical Research: Atmospheres*, 128(7), e2022JD037608.*

*Toride, K., & Hakim, G. J. (2022). What distinguishes MJO events associated with atmospheric rivers?. *Journal of Climate*, 35(18), 6135-6149.*

*Tseng, K. C., Johnson, N. C., Maloney, E. D., Barnes, E. A., & Kapnick, S. B. (2021). Mapping large-scale climate variability to hydrological extremes: An application of the linear inverse model to subseasonal prediction. *Journal of Climate*, 34(11), 4207-4225.*

22. Line 195: Please see comments about lack of interpretation for this entire section.

Response:

Thanks for your suggestions. We added more interpretation in this section in the revised manuscript.

See lines: 197-199. *‘In addition to the influence of climate mode variability on the temporal clustering of unique atmospheric rivers during extended boreal winter, it is also important to consider the seasonality of this temporal clustering [35].’*

See lines: 205-209. *‘The AO mainly impacts Northern California in early winter, while the QBO’s influence is stronger in late winter, increasing unique ARs in the Southwestern U.S., especially California, during its positive phase. In contrast, the QBO influence is relatively small during early winter, which is consistent with the results from Castellano et al. [36].’*

See lines: 215-219. *‘Therefore, this better understanding of how climate modes like AO, QBO, ENSO, MJO, PDO, and PNA influence AR clustering differently in early winter (NDJ) versus late winter (JFM) allows us to tailor AR predictions based on the specific characteristics and behaviors of these climate modes during different parts of the winter season.’*

23. Line 196: “the modulation of unique ARs temporal clustering conditioned upon climate mode variability during boreal winter” is very awkward phrasing, please revise.

Response:

We rephrased this sentence in the revised manuscript.

‘In addition to the influence of climate mode variability on the temporal clustering of unique atmospheric rivers during extended boreal winter, it is also important to consider the seasonality of this temporal clustering.’

24. Why does Fig 4, and only Fig 4, have the absolute value composite plots in addition to the anomaly plots? What critical information does this provide?

Response:

Thanks for your suggestions. We added both of them for MJO plots since the absolute value composite plots in Figure 4(a), alongside the anomaly plots, provide additional critical information by showing the overall magnitude of number of ARs based on different MJO phases, which may not be familiar to audiences. While anomaly plots highlight deviations from a climatological mean (positive or negative), absolute value plots emphasize the total occurrence on the control of MJO.

This dual representation can help distinguish between areas of significant anomaly and areas with generally high or low AR activity, giving a more comprehensive picture of AR distribution and variability across different MJO phases.

To make all composite analysis figures consistent, we moved Figure 4(a) to the Supplementary.

Please see Figure 4 in the revised manuscript.

25. Why does Fig 6 have units of % when Figs 3, 4, 5, 7, and 8 are all in units of unique ARs?

Response:

In Figure 6, we wanted to show how many percentages of AR numbers changed based on MJO phases for early and late winter, which is easier to connect to previous studies about the modulation of MJO on AR frequency seasonality. For instance, Wang et al. (2023) indicate modulation of MJO on AR frequency in JFM in %, and combined with our study, during MJO 6&7 phases, there are significantly higher AR frequencies (~10%) and more unique ARs (10-20%) in the southwestern U.S., while less AR frequency and significantly less unique ARs in the northwestern U.S. Thank you!

26. Figs 7-8: These are differences, not anomalies, so it would be nice to have a different color scale.

Response:

Thanks for your suggestions. We changed the colormap in Figures 7-8 in the revised manuscript.

27. You are inconsistent in the way you describe orientation: sometimes using a meteorological convention (ex. “south-southwesterly”), sometimes using “from X to Y”, sometimes using geographic landmarks (ex. “angled towards the Pacific Northwest”), sometimes using degrees, etc. This is an issue throughout the results + in the introduction and makes it hard to draw useful and informative comparisons.

Response:

Thanks for your suggestion! In addition to using degrees in results, we added meteorological conventions (ex. “south-southwesterly”) to describe orientation in results and the introduction to make them easy to compare. We rephrased this section and the introduction in the revised manuscript.

See lines: 84-92. *‘Slinskey et al. [22] found that AR integrated water vapor transport (IVT) along the U.S. West Coast typically has a southwesterly orientation, aligning with moisture pathways from the subtropics to the extratropics. Guirguis et al. [23] demonstrated that persistent southerly IVT orientations contributed to the exceptionally wet winter of 2016–2017 in Northern California, linking these ARs to significant rainfall. South-southwesterly ARs are associated with more precipitation and stronger moisture transport in Northern California, while westerly orientations are linked to extreme streamflow and intense precipitation in Washington and Oregon, influenced by complex topography [24] [6] [25]. In other regions, west-southwesterly orientations yield the most pronounced hydrological responses in Britain’s Dyfi catchment [26].’*

See lines: 269-290. *‘The bottom row in Figure 9(b) shows the AR IVT orientation climatology (1982-2021) of all AR days in the early and late winter, and Figure 9 (c) bottom row shows the difference between them. AR orientation is defined by the direction of mean IVT, measured clockwise from North (e.g., 0° indicates South to North, 90° indicates West to East) (Figure 9 (e)). Therefore, in Figure 9 (c), a positive anomaly means a more westerly AR IVT, while a negative anomaly indicates a more southerly IVT. In general, the orientation of all ARs days increases from the Northwestern U.S. (~58°-59°, south-southwesterly) to Southern California/Arizona (~67°-68°, west-southwesterly), with a more noticeable shift in late winter. This aligns with Slinskey et al. [22]. The AR orientation difference between NDJ and JFM indicates that the IVT orientation of all AR days is significantly more westerly during early winter than late winter over much of the Western U.S. by around 2°-6° (Figure 9 (c) bottom panel). This shift could be attributed to the seasonal variation in horizontal moisture transport from the subtropics to the extratropics [46, 47]. Late winter (especially January and February) is often characterized by strong southerly winds due to the southernmost locations of the jet stream, leading to a more southerly orientation of ARs [48]. Understanding these differences is crucial for accurately forecasting the hydrological impacts of ARs across different times of the winter and this provides new insights into AR orientation seasonal disaggregation in the western U.S. Based on the study of Picard and Mass [49], precipitation amounts over drainages in the Pacific Northwest can vary substantially with ~10° changes in the direction of AR-related wind flow influenced by the interaction with surrounding orography. Consequently, the difference in AR orientation between NDJ and JFM can result in significant variations in precipitation amounts over the western U.S.’*

28. Line 274: How do your results compare to previous literature about AR orientation in the western US? Has anyone else done a seasonal disaggregation before? If not, it is worth highlighting that as a novel contribution.

Response:

We found no previous literature has done this seasonal disaggregation before, and we highlighted this novel contribution in the revised manuscript. Thank you.

'The AR orientation difference between NDJ and JFM indicates that the IVT orientation of all AR days is significantly more westerly during early winter than late winter over much of the Western U.S. by around 2°-6° (Figure 9 (c) bottom panel). This shift could be attributed to the seasonal variation in horizontal moisture transport from the subtropics to the extratropics [46, 47]. Late winter (especially January and February) is often characterized by strong southerly winds due to the southernmost locations of the jet stream, leading to a more southerly orientation of ARs [48]. Understanding these differences is crucial for accurately forecasting the hydrological impacts of ARs across different times of the winter and this provides new insights into AR orientation seasonal disaggregation in the western U.S.'

29. Line 277-279: You are very close to telling me why these results matter! But you don't finish the thought, which leaves this sentence feeling out of place.

Response:

Thanks for your suggestion! We added more explanations in the revised manuscript.

'Based on the study of Picard and Mass [49], precipitation amounts over drainages in the Pacific Northwest can vary substantially with ~10° changes in the direction of AR-related wind flow influenced by the interaction with surrounding orography. Consequently, the difference in AR orientation between NDJ and JFM can result in significant variations in precipitation amounts over the western U.S.'

30. Some suggestions for Fig 9:

a. 9b bears no relation to 9a and should be a standalone figure.

Response:

Thank you. In Figure 9(b), we want to clarify the definition of All AR days and Clustered Unique ARs we plot in Figure 9(a). Since there are figure number limitations, we reformatted Figure 9 and highlighted the relationship between the two figures in the content and caption.

b. The rightmost column in 9a should be labeled as its own subplot.

Response:

Thanks for your suggestion. We added it as a subplot and revised Figure 9 in the revised manuscript.

c. Instead of the 60 degrees inset on the side of 9a, can you show orientation in terms of the yellow-green-blue color bar? I'm imagining a color fan or a series of colored arrows that allow the reader to easily visually compare, say, 57 vs. 67 degrees.

Response:

Thank you for your suggestion! We revised the 60-degree arrow to a series of colored arrows to facilitate easier comparison for the readers.

d. Another subplot between the current 9a and 9b could show the (clustered AR days) – (all AR days) anomaly, which is arguably more interesting than the NDJ-JFM anomaly and would give visual support to the statement at Lines 287-289.

Response:

Thanks for your suggestion. We added (clustered AR days) – (all AR days) anomaly as a new subplot in the revised manuscript. We also highlighted that they considered different AR IVT; therefore, they may not be apples to apples: we used Life-cycle mean IVT orientation for temporal clustered unique ARs and used daily IVT orientation for all AR days. Therefore, In the case of all AR days, each AR day gets an equal weight when averaging. In the case of clustered unique ARs, a long-lasting AR would get the same weight as a short-lasting AR.

Please see Figure 9 in the revised manuscript.

31. Figs 9a & 10: can you tell me in the figure caption (and in the text) which orientation anomaly is positive and which is negative? Maybe a clockwise/counterclockwise arrow, another

inset to the right of 9a, or text stating that a positive anomaly is more northerly vs. negative is more westerly, etc. Right now I have no idea which is which.

Response:

Thank you. In Figure 9a, a positive anomaly means a larger angle clockwise; therefore, a positive anomaly indicates AR IVT is more westerly. On the opposite, a negative anomaly means a smaller angle clockwise, so a negative anomaly indicates AR IVT is more southerly.

Similarly, in Figure 10, a positive anomaly means a larger angle clockwise, so AR IVT is more westerly than climatology. On the opposite, a negative anomaly means a smaller angle clockwise, so AR IVT is more southerly than climatology.

We added more text in the revised manuscript. See lines: 271-274.

32. Line 366-368: You could say “we consider six dominant climate modes at the daily scale” to avoid the redundancy of specifying that every index is daily. Also, what makes these climate modes dominant?

Response:

We rephrased this sentence in the revised manuscript. Thank you.

We consider these climate modes dominant based on the literature review of their significant impacts on atmospheric rivers and precipitation over the western U.S. and on their utilization in the subseasonal-to-seasonal operational forecasts.

For instance, Guan and Waliser (2015) found that the AO, ENSO, MJO, and PNA significantly impact AR frequency and precipitation in the western U.S. Specifically, the increased AR activity over California was linked to the negative phase of both the PNA and AO. ENSO modulates AR frequency along the western U.S. coast. The AR activity in the California Sierra Nevada is shown to be significantly strengthened when MJO convection is in phase 6 (Guan et al. 2012). The PNA-like pattern strongly correlates with temporal AR clustering in the western U.S. (Zhou et al. 2024). In the operational forecast, MJO and QBO strongly modulate subseasonal AR activity and precipitation in California, with reduced AR activity during easterly QBO and increased activity during westerly QBO in the mid-to-late winter (Castellano et al., 2023). Additionally, Gershunov et al. (2017) reported that AR activity along the west coast of North America generally increases during positive phases of the PDO and decreases during negative phases.

We also added this content in the revised manuscript.

Guan, B., & Waliser, D. E. (2015). Detection of atmospheric rivers: Evaluation and application of an algorithm for global studies. Journal of Geophysical Research: Atmospheres, 120(24), 12514-12535.

Guan, B., Waliser, D. E., Molotch, N. P., Fetzer, E. J., & Neiman, P. J. (2012). Does the Madden-Julian oscillation influence wintertime atmospheric rivers and snowpack in the Sierra Nevada?. Monthly Weather Review, 140(2), 325-342.

- Zhou, Y., Wehner, M., & Collins, W. (2024). Back-to-back high category atmospheric river landfalls occur more often on the west coast of the United States. *Communications Earth & Environment*, 5(1), 187.
- Castellano, C. M., DeFlorio, M. J., Gibson, P. B., Delle Monache, L., Kalansky, J. F., Wang, J., ... & Anderson, M. L. (2023). Development of a statistical subseasonal forecast tool to predict California atmospheric rivers and precipitation based on MJO and QBO activity. *Journal of Geophysical Research: Atmospheres*, 128(6), e2022JD037360.
- Gershunov, A., Shulgina, T., Ralph, F. M., Lavers, D. A., & Rutz, J. J. (2017). Assessing the climate-scale variability of atmospheric rivers affecting western North America. *Geophysical Research Letters*, 44(15), 7900-7908.

33. Line 374: Link does not open, 404 error.

Response:

Thank you for pointing it out. We updated the link to the website and added similar examples of data usage.

'The daily AO and PNA indices are downloaded from the NOAA Climate Prediction Center (CPC), covering January 1950 to the present (<https://www.cpc.ncep.noaa.gov/>; or <https://ftp.cpc.ncep.noaa.gov/cwlinks/>). Similar examples of data usage include Ge and Luo 2023(a)(b).'

*Ge, Y., & Luo, D. (2023)a. Impacts of the different types of El Niño and PDO on the winter sub-seasonal North American zonal temperature dipole via the variability of positive PNA events. *Climate Dynamics*, 60(5), 1397-1413;*

*Ge, Y., & Luo, D. (2023)b. Winter cold extremes over the eastern North America: Pacific origins of interannual-to-decadal variability. *Environmental Research Letters*, 18(5), 054006.'*

34. Line 374-375: Why is ENSO signal removed from PNA, and why is it not removed from any other indices?

Response:

We removed the ENSO signal from PNA since PNA is strongly related to phases of ENSO; projection highly depends on ENSO. For instance, during El Niño events, the PNA tends to exhibit a positive phase, while during La Niña events, it tends to exhibit a negative phase (e.g., Hu et al. 2023; Mueller and Roeckner 2006). This strong dependency can obscure the independent variability and impacts of the PNA. By removing the ENSO signal, we aim to isolate the intrinsic behavior of the PNA. In contrast, other indices may not exhibit such a direct and significant relationship with ENSO, making the removal of the ENSO signal less critical for their independent analysis.

We added more text to clarify in the revised manuscript. Thank you.

'We use linear regression to remove the influence of the ENSO signal on the PNA since PNA is strongly related to phases of ENSO, projection highly depends on ENSO. For instance, during El

Niño events, the PNA tends to exhibit a positive phase, while during La Niña events, it tends to exhibit a negative phase (e.g., Hu et al., 2023; Mueller and Roeckner 2006).'

Hu, S., Zhang, W., Jin, F. F., Hong, L. C., Jiang, F., & Stuecker, M. F. (2023). Seasonal dependence of the Pacific–North American teleconnection associated with ENSO and its interaction with the annual cycle. Journal of Climate, 36(20), 7061-7072.

Mueller, W. A., & Roeckner, E. (2006). ENSO impact on midlatitude circulation patterns in future climate change projections. Geophysical Research Letters, 33(5).

35. Line 386-389: It's not clear what "these climate modes of variability" and "these climate indexes" are referring to. Also the MERRA-2 AR events dataset has not been discussed yet. It seems odd to have this sentence occur before all climate mode datasets have been introduced and to have a paragraph break separating PDO from all other indices.

Response:

We rephrased this sentence, added pointing about the MERRA-2 AR events dataset, and improved the structure in the revised manuscript.

As for PDO, we put it in a separate paragraph since we want to separate short- vs. long-term climate modes and use two different AR datasets based on the constraints of the data timeframe. We clarified this in the revised manuscript.

We also mentioned this in the following question. Thank you!

36. Line 387 & 397: Consider defining the time overlaps in terms of water year (starting in October) rather than the 1940/1941 notation used here for clarity.

Response:

Thank you for your suggestion. We revised the notation to use the water year starting in October in the revised manuscript.

'To consider these climate modes of variability and find the largest time overlap of these climate indexes and the MERRA-2 AR events dataset, we use the period of water years 1983-2021 extended winter from November to March, which is the season with the highest climatological frequency of AR events in the Western U.S.'

'To find the largest time overlap between the PDO index and the longest period of the AR events dataset based on ECMWF Reanalysis v5 (ERA5), we use the period of water years 1941-2018 winter from November to March.'

37. Line 398-401: Okay, now I understand why PDO is in a separate paragraph. It would be

much better to have this information at the start of this section, maybe at Line 369. It would also be helpful to have a pointer to the next section, where you explain why you are using two reanalysis products (i.e., capturing short- vs. long-term climate variability)

Response:

Thank you for your suggestion. We moved this information to the start of the section and improved the structure in the revised manuscript. We also added pointing about the MERRA-2 AR events dataset and the reason of using two reanalysis products.

38. Line 405-407: Sentence fragment

Response:

Thank you. We rephrased this paragraph combined with Reviewer 2's suggestions in the revised manuscript.

39. Line 416: MERRA-2 is at a 3-hour timestep, not 6 hours. ERA5 is at an hourly timestep, not 6 hours.

Response:

Thank you for pointing it out. Here, we want to mention the MERRA-2-based AR catalog we downloaded from Global Atmospheric Rivers Dataverse and the ERA5-based AR catalog we downloaded from the same website, which uses different timesteps with original MERRA-2 and ERA5 datasets. We rephrased this sentence in the manuscript to enhance clarity.

'One is the commonly used MERRA-2-based AR catalog downloaded from Global Atmospheric Rivers Dataverse (<https://dataverse.ucla.edu/dataverse/ar>). This AR dataset covers from 1980 to 2021, with 6-hourly global data with a resolution of $0.5^{\circ} \times 0.625^{\circ}$.'

'Another is the newly published long-period ERA5-based AR catalog downloaded from the same website (<https://dataverse.ucla.edu/dataverse/ar>). This AR dataset goes from 1940 to 2022, with 6-hourly global data and a resolution of $0.25^{\circ} \times 0.25^{\circ}$.'

40. Line 423-434: "long-period climate modes variability PDO" feels awkward. Is there a missing/extra word in here?

Response:

The phrase "long-period climate modes variability PDO" could be clarified as climate models characterized by interdecadal variability such as the PDO. We rephrased this sentence in the manuscript to enhance clarity. Thank you.

'We utilize the ERA5-based dataset to capture the impacts of climate models characterized by interdecadal variability, such as the PDO, on unique ARs (i.e., PDO~ ERA5 AR).'

41. The Guan & Waliser algorithm does not inherently have object IDs, right? How long does

the gap of non-AR timesteps have to be to identify a unique AR event? Can multiple unique AR events occur on the same day? Why do you even need to calculate AR days, when all of the results in Figs 3-8 are measured in terms of unique AR events?

Response:

The dataset provided by Guan via the Global Atmospheric Rivers Dataverse (<https://dataverse.ucla.edu/dataverse/ar>) has AR track IDs to identify unique AR events (Guan and Waliser 2019).

The updated version of the Guan & Waliser algorithm (which is used here) does indeed have object IDs (track IDs in their terminology). A unique AR event (identified with a unique track ID within the entire database) is a temporal sequence of AR objects that spatially overlap between any two adjacent time steps, without temporal gaps. Specifically, they construct AR tracks using 1) Spatial Overlapping: AR shapes at two adjacent time steps are considered part of the same track if they spatially overlap; 2) Temporal Sampling Interval: A 6-hour interval is used, ensuring AR shapes from the same track overlap between adjacent time steps; 3) Shape Evolution: AR shapes are allowed to evolve in size and shape over time. Cases where the AR shape changes significantly are treated as normal track continuations. 4) Continuation: If the AR shape at the current time step overlaps with the AR shape from the previous time step, it is considered a continuation of the track. 5) Genesis and Termination: If the AR shape at the current time step does not overlap with any AR shape from the previous time step, it is considered a new genesis, and the previous AR shape is considered terminated. 6) Complex Situations: These include separation (one AR shape splitting into multiple shapes) and merger (multiple AR shapes merging into one shape).

It is very rare multiple unique AR events occur on the same day at the same grid cell. During the period of water years 1983-2021, we only have a few days with two multiple unique AR events. To better manage this issue, when multiple unique ARs occur on the same day, only the one that initiated earlier is counted.

As for the AR days, we only use it for the AR orientation section to compare the orientation of clustered ARs and all AR days for Figure 9 and provide a new insight into AR orientation.

We added more information, rephrased the sentences, and clarified this in the revised manuscript. Thank you!

Guan, B., and D. E. Waliser (2019), Tracking atmospheric rivers globally: Spatial distributions and temporal evolution of life cycle characteristics, J. Geophys. Res. Atmos., 124, 12523–12552, doi:10.1029/2019JD031205.

42. Line 446: The variability does not alternate between positive and negative phases, the modes themselves do. “Climate modes of variability” is not the same as “variability of climate modes.”

Response:

Thank you for pointing it out. We rephrased this sentence as ‘The climate modes of variability Z_{iq} , which alternates between positive and negative phases, dictates the active and quiet periods of unique ARs’ in the manuscript.

43. Lines 450-454: I have several questions about this methodological choice. When you say “all possible combinations,” what do you mean? Did you include interaction effects? To what degree? Did some coefficients get left out of the model? What regression forms were tested and which were rejected? What is the form of the final regression model?

Response:

Here, to consider the most significant covariates in each grid cell, we cover all possible combinations (i.e., 32 combinations in total, calculated by permutations and combinations $C(5,0)+C(5,1)+C(5,2)+C(5,3)+C(5,4)+C(5,5)$) of AO, QBO, ENSO, MJO, and PNA (Yang, Z., & Villarini, G. (2021)). For instance, the combined model includes combinations like:

~AO
~AO+QBO
~AO+ENSO
~QBO+ENSO+MJO
~AO+QBO+ENSO+MJO

...

Note that the final regression model may be different in different grid cells.

As for the interaction effect, here we removed the ENSO signal from PNA since PNA is strongly related to phases of ENSO, and projection highly depends on ENSO. For instance, during El Niño events, the PNA tends to exhibit a positive phase, while during La Niña events, it tends to exhibit a negative phase (e.g., Hu et al. 2023; Mueller and Roeckner 2006). This strong dependency can obscure the independent variability and impacts of the PNA. By removing the ENSO signal, we aim to isolate the intrinsic behavior of the PNA. Other indices may not exhibit such a direct and significant relationship with ENSO, making the removal of the ENSO signal less critical for their independent analysis. Therefore, since we already remove dependent signals, we do not consider their interaction effect here.

We also took into account the correlation among the different predictors in the context of collinearity. Anderson and Burnham (2004) recommended using a cutoff value of $|0.95|$ for excluding predictors while retaining those with seemingly significant correlations. In this study, the correlation values among these AO, PNA(after removing ENSO), MJO, QBO, and ENSO are generally much smaller than $|0.3|$. Therefore, we do not believe that collinearity among the climate indices has impacted our results.

Yang, Z., & Villarini, G. (2021). *Evaluation of the capability of global climate models in reproducing the temporal clustering in heavy precipitation over Europe. International Journal of Climatology, 41(1), 131-145.*

Anderson, D. and Burnham, K. (2004) *Model Selection and Multimodel Inference, 2nd edition. NY: Springer-Verlag, p. 63.*

* The correlation coefficients of all pairs of AO, PNA(after remove ENSO),MJO,QBO,ENSO:

Please see Table S1

Climate Modes	NDJ	JFM	NDJFM
AO,PNA	-0.2001486018273725	-0.24637864819105046	-0.22586641135570698
AO,RMM1	0.1209974433901579	0.13668287155888773	0.12040836802640792
AO,RMM2	-0.14273152700488279	-0.09281873451330258	-0.11016684247253844
AO,QBO	0.09406993774545983	-0.002884660187138659	0.04219392930520683
AO,ENSO	0.04970780835261645	-0.05224780847040277	0.017161228297032597
PNA,RMM1	-0.12115710713056552	-0.21255015548629774	-0.14734450058746087
PNA,RMM2	0.011772280426781672	0.06678823179269228	0.06126415754558458
PNA,QBO	-0.004741184607769526	0.04540358271568377	0.04404277598393147
PNA,ENSO	-6.121725906730966x10 ⁻⁵	0.0012393480442218633	0.07669100607194652
RMM1,RMM2	-0.026638671121564098	-0.10424254972073081	-0.08375252047633747
RMM1,QBO	0.06267201328652588	0.0036516165803568298	0.018184691699926075
RMM1,ENSO	0.028306267340458997	-0.011848280923704407	0.0160019803792233
RMM2,QBO	-0.05533665225368805	-0.08861361561655196	-0.06668185623789367
RMM2,ENSO	-0.13531087401849876	-0.01258077309207827	-0.10997883961040632
QBO,ENSO	0.18525855802209815	0.14935683005025127	0.15679703260914

44. Line 454: The package name is “survival”, not “Survival”; case matters in R.

Response:

We updated this in the revised manuscript.

45. Lines 459-464: It is repetitive to have the same information twice, once in parentheses and once in a second sentence. I would keep one or the other, and given the choice I would remove the parentheses and keep the second sentence.

Response:

Thank you. We rephrased these sentences to remove the parentheses and keep the second sentence in the revised manuscript.

46. Line 466: Why would you average orientation over AR days when it could be averaged over AR events? Again, why are AR days calculated at all? It is just as simple and more accurate to count the event starts within a 7-day interval.

Response:

Our objective is to provide a comprehensive analysis of AR orientation seasonal disaggregation by considering both AR days and unique AR events. Each approach offers distinct advantages and together they provide a more nuanced understanding of AR behavior in the western U.S.

Calculating the orientation of AR days allows us to capture the continuous impact of ARs over time, which is crucial for understanding their cumulative effects on weather patterns, water resources, and flood risks. This approach also facilitates comparisons with other studies. For

example, the mean orientation of AR days tends to be more southerly in the northwestern U.S. and more westerly in the southwestern U.S. Additionally, this metric is widely used in climatological studies and operational forecasting, as it helps identify prolonged periods of moisture transport and precipitation.

On the other hand, certain AR orientations, when they occur in succession over a short period (unique AR clusters), can exacerbate flood severity. Therefore, we also address the orientation of clustered unique AR events.

We have rephrased the relevant sentences in the introduction section of the revised manuscript to clarify our rationale for using both metrics and to emphasize the added value of this comprehensive analysis. Thank you for your insightful comments!

47. Line 470: If the one-week intervals are fixed, then the results will be different based on the start day; for example, the third AR in Fig 9b would be in a cluster if the one-week window started two days later. Why was the choice made to define temporal clusters in this way, when multiple other definitions of temporally compound ARs exist, and have you tested the effect of the fixed interval start date on your results?

Response:

We appreciate this point and agree that examining this in further detail would be appropriate. Our choice of a fixed 1-week window to define temporal clusters was somewhat arbitrary, but was also related to our desire to examine subseasonal predictability of temporal AR clusters in future work. Weekly aggregates (or two-week aggregates) of variables in subseasonal predictability studies are commonly used.

To address concerns about the sensitivity of our results to the start date of the 1-week interval, we conducted additional tests by removing the first 2 days from the analysis and then recalculating the clustering for both NDJ and JFM periods. As shown in the following figure, the climatology for NDJ and JFM are similar (Figure 9). The difference between NDJ and JFM in terms of clustered unique AR life-cycle orientation in the following figure is still not significant; therefore, we can see in the original text and here that its seasonal changes are not significant. This indicates that our findings are robust to variations in the interval start date.

We have added Figure S4 in the supplementary.

Please see Figure S4

48. Line 473: If I understand correctly, you are comparing the mean orientation of all ARs to the mean orientation of the subset of temporally clustered ARs. I'm not sure how or why calculating the "mean IVT orientation throughout their life cycle" is related to this. Are you averaging twice, once per cluster and once per grid cell? Could you streamline your methods (and your description) to only average once?

Response:

Yes, we are comparing the mean orientation of all ARs with the mean orientation of temporally clustered unique ARs. To capture the full characteristics of each unique AR, we use the life-cycle mean IVT orientation, considering the entire duration rather than just the first day. This approach allows us to better understand the overall behavior and impact of each AR. While this may involve averaging across different stages, it provides a more comprehensive representation of AR orientation throughout its life cycle.

We agree that they may not be apples to apples: we used Life-cycle mean IVT orientation for temporal clustered unique ARs and used daily IVT orientation for all AR days. Therefore, in the case of all AR days, each AR day gets an equal weight when averaging. In the case of clustered unique ARs, a long-lasting AR would get the same weight as a short-lasting AR. We clarified this point in the revised manuscript.

Furthermore, we added a supplementary figure about Life-cycle mean IVT orientation of unique ARs.

Comparing them, we can see unique ARs are more southerly and all ARs days are more westerly, indicating that long-lasting ARs are more westerly. Similarly, comparing unique ARs and clustered unique ARs, we can see temporal clustered unique ARs are more westerly than unique ARs, indicating that clustered unique ARs are more westerly, especially near the coast. Thank you!

Please see Figure S2

49. Fig 10: What anomalies are we looking at? Is this temporally clustered AR orientation vs. all AR orientation? If so, please state that in the text and in the figure caption.

Response:

The anomalies represent a composite analysis of temporally clustered AR orientation based on the positive and negative phases of climate modes, minus the climatology of temporally clustered AR orientation. We have clarified this in both the text and the figure caption.

50. Lines 295-315: Again, there are a bunch of results without much interpretation. Also why did

you highlight the modes you highlighted? All other plots have gotten discussion of every mode (which may be overkill, but it is the precedent you've set).

Response:

Thank you for your suggestion. We have added more interpretation of the results and included a discussion of every mode to ensure consistency with the precedent set in the other plots.

51. Please list the climate modes in the same order throughout the paper (including the figures and figure captions).

Response:

We updated this in the revised manuscript, including the main content, figures, and figure captions. Thank you.

52. Line 324: Not necessarily true; see comment about results coming from different regressions.

Response:

We agree that, given the separation of the PDO and the other five climate modes in our regressions, the statement may not be accurate. Therefore, we have removed this sentence in the revised manuscript. Thank you.

53. Line 331-339: Tell me why each of these findings matters! “This pattern is interesting/unexpected novel because... This is consistent with/differs from other work in that...”

Response:

Thank you for your suggestion. We added more text in the revised manuscript.

For example, *‘This pattern is interesting since MJO significantly modulates AR clustering with the entire Western U.S., which differs from other works that have not observed such a widespread influence.’*

‘This pattern is interesting; since westerly winds are more conducive to precipitation in the Northwestern U.S., while southerly winds favor precipitation in the Southwestern U.S., ARs in early (late) winter are more likely to lead to the most significant hydrologic impacts in the Northwest (Southwest).’

See lines: 357-359, 364-367.

54. Line 334: Please relate this back to which directions have been found to have the largest hydrologic impacts in the western US.

Response:

Thank you for your suggestion. We added more text in the revised manuscript.

'This pattern is interesting; since westerly winds are more conducive to precipitation in the Northwestern U.S., while southerly winds favor precipitation in the Southwestern U.S., ARs in early (late) winter are more likely to lead to the most significant hydrologic impacts in the Northwest (Southwest).'

55. Line 342-343: Why is MJO highlighted again here?

Response:

To put results together and make them clear, we moved this sentence about MJO phases to finding (3) in the revised manuscript. Thank you!

56. Line 347-349: Thank you for connecting your results to previous work. I would love to see more of this throughout the results and conclusion.

Response:

We added more previous results connected to this study in the revised manuscript. Thank you!

57. Lines 340-362: I suggest (a) combining the paragraphs at Lines 340 and 344 and (b) separating the discussion of AR orientation starting at Line 350 from the discussion of forecasting starting at Line 355. The last section of text makes an important point about the value of this work (although it never ties in seasonality), but needs a bit more filling out to be a really strong ending.

Response:

Thank you for your suggestions. We have reorganized these paragraphs based on suggestions and added more text to support the ending.

'Furthermore, the modulation of life-cycle IVT orientation of temporally clustered unique ARs by different climate modes provides us with new ideas for forecasting AR-related precipitation. Specifically, since the impacts of ARs and the related precipitation are closely related to their orientation, climate modes that do not strongly influence temporal AR clustering or frequency can strongly modulate AR precipitation through changes in IVT orientation. For instance, ENSO does not show the influence of AR temporal clustering in the Pacific Northwest but significantly makes the life-cycle IVT orientation of temporally clustered ARs more west-southwesterly during the El Nino phase and more south-southwesterly during the La Nina phase during early winter. Therefore, it will be equally important to consider the impact of AR orientation and temporal AR clustering, frequency, and magnitude in future research. This comprehensive analysis will enhance our understanding of AR impacts under climate modes' modulation with the goal of informing future improvements in precipitation forecasts.'

Hu, H., Dominguez, F., Wang, Z., Lavers, D. A., Zhang, G., & Ralph, F. M. (2017). Linking atmospheric river hydrological impacts on the US West Coast to Rossby wave breaking. Journal of Climate, 30(9), 3381-3399.

58. Line 509: It would be far better if the analysis code (or at least a demonstration script reproducing the main figures) was publicly available via Github or another platform. The analysis and results of this paper are not reproducible as is.

Response:

We put a demonstration script reproducing the main figures in Github UCSD CW3E channel. Codes and data are available there. Please refer: <https://github.com/CW3E/ARclustering>

59. General figure comments:

a. Labels and axes are barely legible and should be in a much larger font. Figs 2, 5, 6, 8, 9a, and 10 are particularly bad.

Response:

We updated these figures in the revised manuscript.

b. Please make the significance dots larger as well.

Response:

We updated significance dots in the revised manuscript. Thank you!

Response to Reviewer 2's comments on:

Temporal clustering of unique atmospheric rivers impacting the Western U.S.: seasonality and climate modes modulation

ZHIQI YANG*, MICHAEL J. DEFLORIO, AGNIV SENGUPTA, JIABAO WANG, CHRISTOPHER M. CASTELLANO, ALEXANDER GERSHUNOV, KRISTEN GUIRGUIS, EMILY SLINSKEY, BIN GUAN, LUCA DELLE MONACHE, F. MARTIN RALPH

Reviewer #2 (Remarks to the Author):

Summary:

Clustering of ARs has profound economical implication as is evidenced in the recent literature. AR orientation is another important factor that is tied to the extent of economic damages. The main idea of this manuscript is the early winter versus late winter influence of six climate modes on temporal clustering and orientation of ARs. The result suggest the AR clustering is modulated by climate modes, AO, PNA etc. and that the extent of modulation varies with the progression through the winter season, for example, QBO influence on early winter clustering is nearly negligible, but in late winter period, there is significant of QBO influence on clustering over west coast.

Based on my holistic assessment of the manuscript, several important inferences can be made from the results, which can be critical for future understand and prediction of ARs over the west Coast. However, there are certain aspects of the manuscript where I think the authors can make improvements-- these are given in my comments below. The manuscript has significant potential for advancing science of ARs, thus I would strongly encourage the authors to revise it for acceptance in the Journal.

Response:

We want to thank the Reviewer for his comments and suggestions, which we have addressed point-by-point below.

Major Comments:

1. While the authors provide compelling literature on evidence of AR clustering (most of the references from 6-17), we may add more discussion on the causes of clustering hypothesized in these references. It is necessary to give a clear and state -of -the -art background on the research question, which is related to the causes of AR clustering; apart from lines 66-69, I did not find literature on the AR clustering causes. I would be better to first establish background on the current understanding of the causes of AR clustering and then build on that to present the research question targeted in the manuscript.

Response:

Thank you for your suggestions! We expanded the introduction to include a more comprehensive discussion of the current research on the causes of temporal AR clustering, as

hypothesized in the cited references. Some of references only mentioned phenomenon and impacts of temporal AR clustering, and we added more text in the second paragraph to highlight current understanding of causing. This additional context helps to establish a clear and state-of-the-art background on the research question addressed in the manuscript.

2. Figure2: Some similarity in the influence of the climate modes can be observed here. For example, notice the -ve effect of AO and +ve effect of PDO over much of the region near the coast. Then one would wonder: are AO and PDO somehow correlated here?

Response:

Thank you for pointing out the similarities in the influence of the AO and PDO observed in Figure 2. The apparent similarity in the influence of the AO and PDO does raise an interesting question about their potential correlation.

While the AO and PDO are distinct climate modes, their impacts can sometimes overlap in certain regions due to the complex interactions within the climate system. Some studies find some lagging relationship between AO and PDO, which is related to Aleutian Low and ocean-atmosphere interaction in the North Pacific (Sun and Wang 2006). However, in this study, the correlation coefficient between AO and PDO during the overlapping period (water year 1982-2018, NDJFM) is relatively low (-0.2). This suggests that while some degree of interaction exists, the correlation is not strong enough to imply a direct or significant relationship within the context of our analysis.

We acknowledge that a more comprehensive statistical analysis could provide deeper insights into the potential interplay between the AO and PDO, particularly concerning their effects on Atmospheric Rivers (AR). This is a promising direction for future research, and we have incorporated additional text in the revised manuscript to explore this potential further.

See lines: 151-152. *'Note that both the PDO and AO significantly impact regions from the North Pacific to the West Coast, warranting further investigation into their interactions in future studies.'*

Sun, J., & Wang, H. (2006). Relationship between Arctic oscillation and Pacific decadal oscillation on decadal timescale. Chinese Science Bulletin, 51, 75-79.

3. L160-177: In Figure 3 and throughout the manuscript, We discuss beta that represents the extend or magnitude of clustering; while as, in Figure 3 and others, we provide number/frequency of ARs as an evidence of clustering. However, clustering extent and frequency of ARs, though related, are two separate concepts. It is possible to have high number of ARs but no clustering, since the ARs may have occurred at regular intervals. I think the authors may try to adopt some better alternative for showing evidence of clustering (Figure 3 and

other). One way would be to consider time between events or its variance as a measure of clustering extent.

Response:

Thank you for pointing it out. We acknowledge that clustering extent and frequency are indeed related but distinct concepts. However, we would like to clarify that the measure we used, the beta (β) coefficient in Cox regression, directly indicates whether events exhibit temporal clustering. When beta (β) is statistically significant, it signifies that events are temporally clustered under the influence of a specific climate mode (Z), as explained by the nature of the Cox regression equation (Villarini et al., 2013).

To address this, we use Figure 2 to illustrate the clustering extent, while Figure 3 and other composite analyses are intended to support our findings from a different angle, highlighting modulation patterns based on climate modes and the changes in number.

Villarini, G., et al., On the temporal clustering of US floods and its relationship to climate teleconnection patterns. International Journal of Climatology, 2013. 33(3): p. 629-640.

4. Figure 9a: Comparing top and bottom panels, we can see that unique clustered ARs are more southerly than all AR days; we may note this and try to discuss what may be the cause. Southerly orientation of unique clustered ARs and westerly orientation of all ARs days indicate that over their lifetime, ARs shift from southerly to westerly. This implication is based on the definition of unique clustered ARs shown in Figure 9b. We may think about this.

Response:

Thank you for your insightful observation regarding Figure 9a. To understand why the unique clustered ARs tend to have a more southerly orientation compared to the overall AR days, we added a supplementary figure about Life-cycle mean IVT orientation of unique ARs (supplementary Figure S2).

Since we used Life-cycle mean IVT orientation for temporal clustered unique ARs and unique ARs, while used daily IVT orientation for all AR days, the weights of each AR are different in these cases. Specifically, in the case of all AR days, each AR day gets an equal weight when averaging, indicating long-lasting AR would get a higher weight. In the case of clustered unique ARs or unique ARs, a long-lasting AR would get the same weight as a short-lasting AR.

Comparing them, we can see unique ARs are more southerly and all ARs days are more westerly, indicating that long-lasting ARs are more westerly. Similarly, comparing unique ARs and clustered unique ARs, we can see temporal clustered unique ARs are more westerly than unique ARs, indicating that clustered unique ARs are more westerly, especially near the coast.

We added more text to discuss it in the revised manuscript.

See lines: 301-305. *The temporal clustered unique ARs are more southerly, and all ARs days are more westerly, indicating that long-lasting ARs are more westerly since in the case of*

all AR days, a long-lasting AR would get a higher weight than a short-lasting AR. Similarly, comparing unique ARs (see Methods; Figure S2) and clustered unique ARs, we can see temporal clustered unique ARs are more westerly than unique ARs, especially near the coast.'

Please see Figure S2

5. L439: Isn't this Poisson process for each day? If yes, then what is point of naming it as Cox process? Writing the complete probability density function here would be useful for better interpretation. why does t range from 1 to 365 when we only consider winter AR?

Response:

It's important to clarify that we are using a Cox process, not a standard Poisson process. A Poisson process assumes that events occur independently at a constant rate over time and the rate of occurrence λ is time-independent, which means that the occurrence of one event does not change the probability of occurrence of a subsequent one and it bears no information about other events.

However, a Cox process (also known as a doubly stochastic Poisson process) allows the intensity function to be a random process itself, the rate of occurrence parameter λ is not deterministic (either constant as in homogeneous Poisson processes or a deterministic function of time as in inhomogeneous Poisson processes), but it is a random function (λ varies over time).

The use of the Cox process in the temporal clustering of hydrometeorology variables can be found in Smith and Karr 1985, Villarini et al. 2013, and Mallakpour et al. 2017. We added more explanation in the revised manuscript.

Here are probability density functions of them, and we can find more details here Villarini et al. 2013:

For a **Poisson process**, the probability that exactly k events occur in a fixed time interval t is given by the Poisson distribution:

$$P(N(t) = k) = \frac{(\lambda t)^k e^{-\lambda t}}{k!}$$

where:

- λ is the rate of occurrence (constant),
- t is the time interval,
- k is the number of events,
- e is the base of the natural logarithm.

On the other hand, for a **Cox process**, conditionally on the realization of the stochastic intensity function $\lambda(t)$, the number of events $N(t)$ in the time interval $[0, t]$ follows a Poisson distribution:

$$P(N(t) = k | \lambda(t)) = \frac{(\Lambda(t))^k e^{-\Lambda(t)}}{k!}$$

$$\Lambda(t) = \int_0^t \lambda(s) ds$$

Here we use the t^{th} day ($t=1,2,\dots,151$) in the i^{th} year

$$\lambda_i(t) = \lambda_0(t) \exp\left[\sum_{q=1}^m \beta_q Z_{iq}(t)\right]$$

Thanks for pointing out the t range. We revised t range as from 1 to 151 for NDJFM, t range as from 1 to 92 for NDJ, and t range as 1 to 90 for JFM since we consider only winter.

Smith JA, Karr AF (1985) Statistical inference for point process models of rainfall. Water Resour Res 21(1):73–79

Villarini G, Smith JA, Vitolo R, Stephenson DB (2013) On the temporal clustering of US floods and its relationship to climate teleconnection patterns. Int J Climatol 33(3):629–640

Mallakpour I, Villarini G, Jones MP, Smith JA (2017) On the use of Cox regression to examine the temporal clustering of flooding and heavy precipitation across the central United States. Global Planet Change 155:98–108

Minor Comments:

Figure3: In regions where frequency is nearly zero, how did the authors tackle the most-often zero issue?

Response:

We addressed the most-often zero issue by aggregating the unique AR counts within a one-week window. This approach helps to mitigate the impact of zero values in individual grid cells. By using a one-week aggregation, the likelihood of having non-zero AR counts increases, thus reducing the sparsity of the data and ensuring more reliable computation of anomalies. This method effectively smooths out the data and provides a more robust basis for statistical analysis and anomaly detection. Thank you!

Figures 4&5: What is the underlying null hypothesis at each gridcell for which statistical significance of estimated?

Response:

The underlying null hypothesis at each grid cell in Figures 4 and 5 is that there is no significant difference in the number of unique ARs in a 1-week window during the winter season (NDJFM in Figure 4, NDJ or JFM in Figure 5) between the composite analysis based on positive or negative phases of climate modes (or combined MJO phases in Figure 4) and the long-term average from 1982/1983 to 2020/2021. In other words, it posits that any observed anomalies in the number of unique ARs at each grid cell are due to random variability rather than an effect of the climate modes phases. Statistical significance at the 5% level indicates that there is less than a 5% probability that the observed anomalies are due to random chance.

We added more text in Methods section.

Figures3 &7 seem redundant; we may keep Figure 7 only.

Response:

In Figure 3, we show Composite analysis in anomalies of the number of unique ARs in the 1-week window during NDJFM (entire winter), while in Figure 7, we show the difference between NDJ and JFM, i.e., NDJ minus JFM. In Figure 7, we can directly see the seasonality of climate modes modulation on number of unique ARs in the 1-week window. We revised the figure captions to clarify this. Thank you.

L255-260: Again Figures 7 &8 seem redundant.

Response:

We included the impacts of MJO indexes (RMM1/RMM2) and MJO four combined phases (phases 8&1, 2&3, 4&5, 6&7) because they are both commonly used in understanding and forecasting S2S AR in previous studies and operational forecasts.

In Figure 7, we show the difference between NDJ and JFM in terms of AO, QBO, ENSO, RMM1/RMM2, PDO, and PNA, while in Figure 8, we focused on MJO phases 8&1, 2&3, 4&5, 6&7. We want to use Figures 7 and 8 to show the difference more intuitively. Thank you for pointing it out.

L45: All of the 1-5 references are not on flood damages, please verify.

Response:

Thanks for your suggestion. References 1 mentioned Atmospheric rivers drive flood damages, and others mentioned Atmospheric rivers drive precipitations and flood risks. We rephrased this sentence in the revised manuscript to make them match.

*‘Specifically, ARs significantly contribute to **flood risks** and precipitation in the Western U.S. [1-3], and flood damages exhibit exponential growth with the intensity and duration of AR occurrences [1].’*

L46: Again, all of the 6-9 references do not appear related to the topic of discussion; reference 10 is not on ARs.

Response:

We updated these references in the revised manuscript. Thank you!

‘When multiple ARs occur in rapid succession, the compound effect on the hydrologic system can lead to more hazards (Bowers et al. 2023), more impacts on the ecosystem [5], and more economic losses [1].’

Bowers, C., Serafin, K. A., Tseng, K. C., & Baker, J. W. (2023). Atmospheric River Sequences as Indicators of Hydrologic Hazard in Present and Future Climates. Authorea Preprints.

L47: please elaborate what is meant by indicator here.

Response:

The ‘AR sequences can also be indicators’ means that AR sequences are aligned with periods of increased hydrologic hazards in the historical record.

We rephrased this sentence in the revised manuscript to make it clear. Thank you.

‘AR sequences are aligned with periods of increased hydrologic hazards in the historical record, such as extreme precipitation, landslides, and flooding (Bowers et al. 2023).’

Bowers, C., Serafin, K. A., Tseng, K. C., & Baker, J. W. (2023). Atmospheric river sequences as indicators of hydrologic hazard in historical reanalysis and GFDL SPEAR future climate projections. Earth's Future, 11(12), e2023EF003536.

L58-59: if they occur in bursts, the by definition they are dependent, right?

Response:

Yes, here we want to say unique ARs; each of them has different track IDs. And in statistics they are dependent when they occur in bursts. We rephrased this sentence in the revised manuscript to clarify this. Thank you!

L70: Does this mean rate of AR clustering is more than clustering at random?

Response:

Yes, this indicates that the rate of AR clustering observed is higher than what would be expected from random clustering. We have clarified the term that in the revised manuscript, we mean consider unique AR events (or, in short, 'unique ARs'). Each of them has different track IDs, and they are dependent when they occur in bursts. Thank you!

'Here, we consider unique AR events (or, in short, 'unique ARs'), which means for an AR event, if it occurred on consecutive days, we consider only the first day of the event.'

L82: We may keep either "mean" or "primarily" not both.

Response:

We rephrased this sentence in the revised manuscript to make it clear. Thank you.

'Slinsky et al. [22] found that AR integrated water vapor transport (IVT) along the U.S. West Coast typically has a southwesterly orientation, aligning with moisture pathways from the subtropics to the extratropics.'

L91: This doesn't appear to be linked with relationship between AR orientation and extreme precipitation, which is the main theme of the paragraph.

Response:

Thank you for your suggestions! We rephrased this sentence and combined with other references in the revised manuscript to highlight the relationship between AR orientation and extreme precipitation.

'South-southwesterly ARs are associated with more precipitation and stronger moisture transport in Northern California, while westerly orientations are linked to extreme streamflow and intense precipitation in Washington and Oregon [24] [6] [25].'

We also rephased the first sentence in this paragraph:

Moreover, the orientation of ARs has distinct effects on the intensity and duration of their associated precipitation and floods, with particular emphasis on their orientation upon landfall due the interplay between moisture transport and complex terrain.'

L287-289: I wonder why this may be the case?

Response:

They are significantly different since they are considered different AR IVTs. They may not be apples to apples: we used Life-cycle mean IVT orientation for temporal clustered unique ARs and used daily IVT orientation for all AR days. Therefore, In the case of all AR days, each AR day gets an equal weight when averaging. In the case of clustered unique ARs, a long-lasting AR would get the same weight as a short-lasting AR.

To make it clearer, we also added figure (clustered AR days) – (all AR days) anomaly as a new subplot in the revised manuscript. Thank you!

Please see Figure 9 in the revised manuscript.

L374-375: This is important; please explain in detail.

Response:

Thank you for your suggestion. We removed the ENSO signal from PNA since PNA is strongly related to phases of ENSO; projection highly depends on ENSO. For instance, during El Niño events, the PNA tends to exhibit a positive phase, while during La Niña events, it tends to exhibit a negative phase (e.g., Hu et al. 2023; Mueller and Roeckner 2006). This strong dependency can obscure the independent variability and impacts of the PNA. By removing the ENSO signal, we aim to isolate the intrinsic behavior of the PNA. In contrast, other indices may not exhibit such a direct and significant relationship with ENSO, making the removal of the ENSO signal less critical for their independent analysis.

We added more text to clarify in the revised manuscript. Thank you.

‘We use linear regression to remove the influence of the ENSO signal on the PNA since PNA is strongly related to phases of ENSO, projection highly depends on ENSO. For instance, during El Niño events, the PNA tends to exhibit a positive phase, while during La Niña events, it tends to exhibit a negative phase (e.g., Hu et al., 2023; Mueller and Roeckner 2006).’

*Hu, S., Zhang, W., Jin, F. F., Hong, L. C., Jiang, F., & Stuecker, M. F. (2023). Seasonal dependence of the Pacific–North American teleconnection associated with ENSO and its interaction with the annual cycle. *Journal of Climate*, 36(20), 7061-7072.*

*Mueller, W. A., & Roeckner, E. (2006). ENSO impact on midlatitude circulation patterns in future climate change projections. *Geophysical Research Letters*, 33(5).*

L403-404: At other instances, we say AR catalog from 58, here we are detecting ARs. If we already have an AR catalog, then lines 403-414 may be omitted.

Response:

Thank you for your suggestions. Since we already have AR catalogs, we removed details in lines 403-414 and retained only the name and evaluation of this algorithm.

L426-427: how is independence defined?

Response:

We want to say unique ARs; each of them has different track IDs. We rephrased this sentence as ‘unique ARs’ in the revised manuscript to clarify this term. Thank you!

‘We identify time series of unique ARs that contain occurrence or non-occurrence (represented as ‘1’ or ‘0’) at a daily scale in each grid.’

L435: So the events are not independent, right?

Response:

Thanks for pointing it out. Yes, events are dependent when they occur in bursts. Here we want to say unique ARs; each of them has different track IDs. We rephrased all relevant sentences and clarified the term in the revised manuscript to use the same term, 'unique ARs.'

Dear Editor and Reviewers,

We would like to sincerely thank your thorough second review and insightful comments on our manuscript, which we hope to have improved the quality of this work. Please see our responses below (in blue) to each comment (in black).

Sincerely,

Zhiqi Yang

Response to Reviewer 1's comments on:

Temporal clustering of unique atmospheric rivers impacting the Western U.S.:
seasonality and climate modes modulation

Current title: Seasonality and climate modes influence the temporal
clustering of unique atmospheric rivers in the Western U.S.

ZHIQI YANG*, MICHAEL J. DEFLORIO, AGNIV SENGUPTA, JIABAO WANG, CHRISTOPHER M.
CASTELLANO, ALEXANDER GERSHUNOV, KRISTEN GUIRGUIS, EMILY SLINSKEY, BIN GUAN, LUCA
DELLE MONACHE, F. MARTIN RALPH

Reviewer #1 (Remarks to the Author):

Thank you to the authors for their edits to the manuscript and their thoughtful responses to the comments. I really appreciate the additional interpretive text that has been added throughout, and I like the symmetry now between the way temporal clustering + orientation are discussed in the introduction. I have two remaining items of concern, as well as a few editorial comments below. I believe that this article is ready for publication with minor revisions.

Response:

We want to thank the Reviewer for her/his comments and suggestions, which we have addressed point-by-point below. Additionally, we have revised/rephrased some text to comply with the journal's word limit.

First, Fish et al. (2019, 2022) use a five-day instead of a seven-day window, and they use an algorithm that starts counting a cluster/family based on some threshold criteria rather than using fixed intervals. To be clear, I think all of these choices made in this manuscript are defensible

(and thank you for testing the sensitivity of the start of the fixed window), but it is incorrect to state that the choices are directly supported by the cited literature.

Response:

Thank you for pointing this out. We agree it isn't directly supported by the cited literature, while our methodology was inspired by similar concepts, and we understand that Fish et al. used a five-day window and a threshold-based algorithm to identify clusters, rather than the fixed seven-day intervals used in our study. To better clarify this concept, we have revised the text to clarify that these references serve as contextual examples rather than direct methodological support.

We also appreciate your acknowledgment of our sensitivity testing for the window start.

'Inspired by the clustering framework presented by Fish et al. [10, 11], we implement a 1-week window to identify AR families here. While Fish et al. [10, 11] classify AR events as an AR family if two or more AR objects occur within the 5-day aggregation period, we adapted similar concepts to define AR families. Therefore, 1-week is an appropriate window to identify clustering.'

Second, I do not understand the rationale for comparing clustered AR events against all AR days in Fig 9. As mentioned in both the authors' comment responses (30d) and in the manuscript itself (Line 468), this is not an apples-to-apples comparison, because the orientation of long vs. short clustered ARs is weighted equally but a long AR would be weighted more heavily than a short one in the mean orientation of AR days. The effects of clustering and duration are therefore lumped together with no way to distinguish the contributions of each individually. It would be more consistent to compare either (a) all days within clustered ARs vs. all AR days or (b) clustered unique AR events vs. all unique AR events. Considering that analysis (b) is already done in Fig S2, I believe that it belongs in the main text in place of Fig 9.

Response:

Thank you for your insightful suggestions. Following your suggestion, we agree that comparing clustered unique AR events to all unique events would provide a clearer distinction. Given that analysis (b) is already presented in Fig S2, we moved it to the main text to replace Fig 9 for a more consistent and focused analysis and revising text accordingly. We relocated the comparison of clustered AR events to all AR days in original Fig 9 to the supplementary materials, providing additional context for all AR days without confounding the main analysis.

Please see new Figure 9 and Figure S2

Figure 9. **Panel (a):** Schematic of terms used to describe all AR days, unique ARs, and temporal clustered unique ARs. The temporal clustered unique ARs satisfy the situation where there is more than one unique AR in a week window. **Panel (b):** Life-cycle IVT orientation of temporal clustered unique ARs based on MERRA-2 AR dataset from 1982/1983 to 2020/2021 in early winter (NDJ, first row left), late winter (JFM, first row middle), and their difference (first row right). Life-cycle IVT orientation of unique ARs in early winter (NDJ, second row left), late winter (JFM, second row middle), and their difference (second row right). Life-cycle IVT orientation of temporal clustered unique ARs minus of unique ARs in early winter (NDJ, third row left). Life-cycle IVT orientation of temporal clustered unique ARs minus of unique ARs in late winter (JFM, third row right). Unit: degree. **Panel (c):** IVT orientation diagram.

Additional comments:

- Line 134: run-on sentence

Response:

Thank you for your suggestion. We rephrased this sentence in the revised manuscript.

‘South-southwesterly ARs are associated with more precipitation and stronger moisture transport in Northern California, while westerly orientations are linked to extreme streamflow and intense precipitation in Washington and Oregon.’

- Line 141: great addition

Response:

Thank you, we are glad to hear that.

- Line 193: move “and” to last item in list

Response:

We rephrased this sentence in the revised manuscript.

‘Since climate modes are important predictors for subseasonal-to-seasonal (S2S) forecasting modeling studies, here we consider several major climate modes with important impacts on the Western U.S., including the Arctic Oscillation (AO) [28], quasi-biennial oscillation (QBO) [29], ENSO, MJO [30, 31], Pacific Decadal Oscillation (PDO) [32], and Pacific-North American pattern (PNA) [33, 34].’

- Line 225: should this be “temporal clustering of unique ARs”?

Response:

Thank you for pointing it out. We rephrased this sentence in the revised manuscript.

‘Then, we investigate the temporal clustering of unique ARs within the extended boreal winter over the Western U.S., and the impact of AO, QBO, ENSO, MJO, PDO, and PNA (see Methods) on the magnitude and spatial structure of the clustering.’

- Line 229: it would be helpful to move some the contextual description about what the Cox regression measures, what the coefficient beta measures, etc. from the Methods up to here

Response:

Thank you for your suggestion. We moved some description about what the Cox regression and coefficient beta measures here from Methods.

‘Cox regression coefficients β (shaded areas) indicate that unique ARs exhibit temporal clustering. The Cox process belongs to the family of clustered processes, indicating the occurrence of one event has connections with the subsequent one. β represents the regression coefficient for

each covariate, quantifying the influence of climate modes on the cluster distribution of unique ARs, where higher $|\beta|$ values indicate stronger temporal AR clustering (see Methods).'

- Line 275: should be present tense

Response:

Thank you for your suggestion. We rephrased this sentence in the revised manuscript.

'The spatial patterns in Figure 3 highly match the Cox regression coefficients patterns in Figure 2 left column, supporting that unique ARs exhibit temporal clustering and are affected by climate modes.'

- Fig 9:

- o The labels of 9a and 9b seem to be flipped, please check in-text references to both
- o It would be helpful if Fig 9 were organized in the same order it was described in the text: first the schematic, then the top row of (b), then the bottom row of (b), etc.
- o Why does Fig 9(c) have a different color scale than 9(d) for the same values?
- o Please mention the orientation diagram 9(e) somewhere in text

Response:

Thank you for your suggestions.

We have updated the text and labels for 9a and 9b to ensure they are consistent and organized Fig 9 in the same sequence described in the text. We also replace original Fig 9b for a more consistent and focused analysis and revising main text, accordingly, followed by your main suggestion about using *'(b) clustered unique AR events vs. all unique AR events'*.

To improve clarity, we now introduce all subplots in Fig 9 at the beginning, aligning with discussing AR IVT orientation for all AR days in the first paragraph of this section, following by the orientation of unique ARs and temporally clustered unique ARs in the next paragraph.

'Figure 9(a) shows schematic of terms used to describe all AR days (black squares), unique ARs (light blue), and temporal clustered unique ARs (dark blue). Figure 9(b) shows the life-cycle IVT orientation of temporal clustered unique ARs (top row, left and middle) and all unique ARs (second row, left and middle) during NDJ and JFM. The right-side panels of Figure 9(b) display the NDJ minus JFM differences, while the bottom panels highlight the first-minus-middle rows difference of Figure 9(b). We show AR IVT orientation climatology of all AR days in Figure S2. AR orientation is defined by the direction of mean IVT, measured clockwise from North (e.g., 0° indicates South to North, 90° indicates West to East) (Figure 9 (c)).'

For the color scales, we revised color scales as the same.

Additionally, we have now mentioned the orientation diagram in 9(c) within the text. *'e.g., 0° indicates South to North, 90° indicates West to East) (Figure 9 c)'*

Please see new Figure 9 and Figure S2

- Lines 467 & 471: number supplementary figures

Response:

Thank you. We added supplementary figures number in the revised manuscript.

‘The seasonality of the orientation of temporal clustered unique ARs is significantly different from the seasonality of the orientation for all AR days (Figure 9(b) and Figure S2). Also, comparing unique ARs and clustered unique ARs, we can see temporal clustered unique ARs are more westerly than unique ARs, especially near the coast (Figure 9(b) bottom panels).’

- Line 517 “in opposing ways” – relative to what?

Response:

Thank you for pointing it out. We propose that certain phases of climate modes associated with reduced AR temporal clustering (also reducing AR-related precipitation) or showing fewer impacts on temporal AR clustering may also influence AR IVT orientation, enhancing precipitation. For instance, while ENSO strongly modulates AR orientation, it has a relatively weak impact on AR temporal clustering across the Western U.S. To make it clear, we have rephrased the sentence in the revised manuscript as *‘Therefore, climate modes strongly influence the IVT life-cycle orientation of temporally clustered unique ARs, which in turn affects whether they enhance or reduce precipitation.’* and leave the different impacts of climate modes on IVT orientation and temporal clustering itself such as ENSO in the conclusion section: *‘ENSO shows a strong modulation of the IVT life-cycle orientation of temporally clustered unique ARs, which differs from its weak impacts on temporal AR clustering’.*

- Line 549: ARs? Or clustered ARs?

Response:

It is clustered ARs. We rephrased this sentence in the revised manuscript.

‘This pattern is interesting; since westerly winds are more conducive to precipitation in the Northwestern U.S., while southerly winds favor precipitation in the Southwestern U.S., temporally clustered ARs in early (late) winter are more likely to lead to the most significant hydrologic impacts in the Northwest (Southwest).’

- Line 796: typo

Response:

We edited it in the revised manuscript.

‘We use a 1-week window to measure temporal compounding of ARs in composite analysis, which is to establish a foundational framework for future research on subseasonal AR clustering and prediction within a dynamic statistical forecasting framework.’

- Line 826: typo

Response:

We edited it in the revised manuscript. Thank you.

'Our results are not sensitive to the start date of the one-week interval (Figure S4).'

Response to Reviewer 2's comments on:

Temporal clustering of unique atmospheric rivers impacting the Western U.S.:
seasonality and climate modes modulation

Current title: Seasonality and climate modes influence the temporal
clustering of unique atmospheric rivers in the Western U.S.

ZHIQI YANG*, MICHAEL J. DEFLORIO, AGNIV SENGUPTA, JIABAO WANG, CHRISTOPHER M.
CASTELLANO, ALEXANDER GERSHUNOV, KRISTEN GUIRGUIS, EMILY SLINSKEY, BIN GUAN, LUCA
DELLE MONACHE, F. MARTIN RALPH

Reviewer #2 (Remarks to the Author):

Dear Editor

Sorry for the delayed review. I'm happy with the revision made by the authors.

Thank you

Response:

We sincerely appreciate the reviewer's valuable comments and suggestions, and we are pleased to hear that you are satisfied with the revisions.